# A Two-Way Fixed Effects Estimation on the Impact of Industrial Land Supply on Environmental Pollution in Urban China

**DOI:** 10.3390/ijerph192214890

**Published:** 2022-11-12

**Authors:** Xiangqi Yan, Hanbing Tuo, Yani Lai

**Affiliations:** 1School of Economics, Jinan University, Guangzhou 510632, China; 2School of Civil and Transportation Engineering, Shenzhen University, Shenzhen 518055, China

**Keywords:** ecological environment, industrial land transfers, local government, environmental pollution, China

## Abstract

Despite the great economic growth and fast urbanization process in the past four decades, China is now suffering severely from environmental pollution. Local governments’ industrial land supply behaviors have a great impact on local investment, economic growth, and environmental pollution, which has not been effectively evaluated. To fill this gap, this paper quantitatively investigates the impact of industrial land supply by local governments on environmental pollution based on a two-way fixed effects model. A comprehensive and reliable data set for 277 Chinese prefecture-level cities from 2009 to 2017 has been collected for analysis. The findings suggest that the increase of the ratio of industrial and mining storage land to total land supply significantly increases the concentration of PM2.5. The results remain significant and robust after a series of robustness tests. The negative impacts on environmental quality caused by differences in land supply behavior are greater in the central and western regions. We further explored intermediate mechanisms for the environmental impact of local governments’ allocations of industrial land. The findings suggest that greater industrial land transfer by local governments leads to an expansion in the scale of regional secondary industry and increases in local fiscal deficit. Unbalanced industrial development, insufficient corporate innovation, and insufficient investment in environmental protection will increase pollution. This study provides a reference for improving regulatory measures on land transactions and for formulating regional polices for environmental protection.

## 1. Introduction

Economic growth and environmental pollution are key issues for policy makers and social scientists. The Chinese economy has grown very rapidly in the forty years following its “reform and opening” but at a large environmental cost. The Chinese economy is still at the left half of the environmental Kuznets curve (EKC), so it is crucial to consider both the country’s economic growth and the environmental protection needs [1]. Based on the Bulletin on the State of Chinese Ecological Environment in 2020, only 56.7% of the country’s 337 prefecture-level cities meet government air quality standards and 83.4% of the national surface water sections satisfy the drinking water quality requirements. Therefore, air pollution and water pollution are still essential challenges for environmental protection and sustainable development. On the one hand, environmental pollution severely threatens people’s physical and mental health and carries enormous economic costs. On the other hand, with improvements in economic conditions and living standards, households now expect a higher quality of health and environment and are even willing to pay for this. The Chinese economy is now under urgent pressure to become sustainable and to combine high-quality growth with environmental protection and pollution control. It is also necessary to make efforts to solve the ecological problem from its root.

Existing research [2,3,4] has recognized the important role of the local governments’ land resource allocation behavior on environmental problems. In the context of fiscal decentralization and the “GDP-growth tournament among governments”, local government officials are generally inclined to adopt the model of “developing through land” due to financial and political career promotion incentives [2]. They tend to pursue regional economic growth goals and to maximize fiscal revenue without fully considering regional environmental protection and governance. Specifically, local governments tend to adopt the method of transferring industrial land by agreement and apply a low entry threshold for absorbing investment to attract industrial enterprises to the region and relax regional environmental regulations, which aggravates environmental pollution. In addition, the extensive development model of “land for capital” (selling industrial land cheaply) can quickly increase regional fixed asset investment, cultivate new tax sources, make up the gap between fiscal revenue and expenditure, and improve economic growth. With the advantage of a “two-way monopoly” in land transactions supported by China’s land property rights system [5], local governments expropriate land from rural areas at a price lower than the land’s value and then supply land for real estate, commerce, and service industries at a high price to gain considerable land transfer fees. This differentiated land allocation strategy for industrial, commercial, and residential land is local governments’ optimal strategic choice to maximize their own utility. However, the resulting inefficient supply of industrial land, the damage to the ecological environment, and their regional heterogeneity need to be carefully evaluated; to date this has not happened.

To fill this gap, this study aims to investigate the impact of industrial land supply on environmental pollution in China through the combination of spatial analysis and regression models. We first use GIS to investigate the spatial variation of pollution and industrial land supply. We then apply a two-way fixed effects regression method to quantitatively evaluate the impacts of industrial land supply on environmental pollution. We further investigate the regional heterogeneity rigorously through group-level regressions to see whether the differentiated land allocation strategies adopted by local governments can aggravate regional environmental pollution. To test the robustness, we use alternative explained variables and narrow the samples successively. Further, we explore the intermediate mechanisms from three dimensions through a two-step method, the Sobel test, and the bootstrap test. Lastly, we provide policy implications for optimizing the allocation of land resources and improving land transfer regulations and environmental protection policies. The remainder of this paper is arranged as follows. Section 2 reviews the literature on environmental pollution and land resource allocation. In this section, we first discuss the current situation of environmental pollution in China and then introduce the economy–environmental trade-off for local governments. Then, we analyze the incentives of industrial land supply behavior of local governments and the consequences on the local environment. Section 3 explains our research design, data sources, empirical framework, and the variables used in this study. In Section 4, we first provide spatial distributions of our explanatory and explained variables. Then, we present our empirical results. Section 5 provides further analysis on regional heterogeneity and explores the mechanisms. Section 6 concludes the study and provides policy recommendations.

## 2. Literature Review

Environmental pollution and its influential factors have been important topics of research among scholars around the world. Many researchers attempt to explain the evolution of environmental conditions from the perspective of economic development. Grossman and Krueger [6] found that concentrations of sulfur dioxide and smoke initially increase with a per capita GDP at low levels of national income, then decrease with a higher per capita GDP, and finally level off with a per capita GDP of USD 8000 (1985 dollars). This inverted U-shaped relationship is commonly referred to as an environmental Kuznets curve (EKC). Hettige et al. [7] found a similar inverted U-shaped relationship between GDP and the toxic intensity of manufacturing industries. Much other empirical research has proved the validity of EKC in different regions of the world, including North America [8,9] and South-West Europe [10]. Some research has found similar results in China. For example, Wang and Huang [11] found a U-shaped relationship between economic growth and air pollution in 112 cities in China. Using an ARDL model, Akram et al. [12] found strong evidence of the EKC for China. Shao et al. [13] found a statistically significant U-shaped relationship between the degree of smog pollution and regional economic growth, but that most of the eastern region is still on the left side of the EKC, that is, the environment is gradually deteriorating along with economic development. However, some researchers do not agree with the shape of the EKC. Xu and Song [14] found an inverted U-shaped relationship exists between income and pollution in China as a whole and in the eastern and central regions, although the western region demonstrates a U-shaped pattern. A condition for establishing the EKC’s inverted U-shaped curve is that when economic development reaches a certain stage, measures such as increased investment in environmental protection, improved quality of governance [15], technological progress, improved energy efficiency [16], and the relocation to different places of high-polluting and high-energy industries can curb or alleviate environmental pollution. However, if institutional barriers inhibit the above mechanisms, then a higher level of economic development may not be accompanied by reduced environmental pollution. Therefore, the problem of environmental pollution is partly rooted in the administrative management system, so it is necessary to explore the role of the local governments in environmental pollution.

Under China’s system of decentralization, local governments implement environmental protection policies promulgated by the central government and are also the bridge between the central government and local enterprises [17]. As the designers and implementers of regional policies, local governments’ actions will inevitably affect economic development and environmental conditions in their jurisdictions, which are reflected in the allocation of land resources and discretionary decisions in implementing policies [18]. For a long time, local governments in China have been more concerned about economic growth than environmental pollution. Local officials usually have insufficient incentive to implement environmental supervisions [19] due to their political career promotion incentives and will tend to allocate resources to areas that can promote short-term economic growth. The motivations for local governments’ land transfer behaviors are also inseparable from the financial and political promotion incentives for officials. On one hand, land finance is an important source of income for local governments in China. On the other hand, local officials are motivated to “collaborate” with companies to achieve mutual benefits: the local governments earn large profits from land transfer agreements and reduced environmental regulations and supervision to support high pollution and high energy consumption enterprises. These enterprises attract investment and create long-term benefits such as fixed-asset investment, employment, attraction of talented workers, and financial taxation for the local governments. In 2013, the central government incorporated local environmental protection indicators into the official performance assessment system. However, due to the positive externalities of environmental governance and information asymmetry in the top-down assessment system of central and local principal-agent systems, local government officials still have insufficient incentives for environmental protection. The existing literature argues that “government-enterprise collusion” is one of the reasons for the high level of environmental pollution in China [3,4]. Guo and Yao [20] argued that local governments have neglected environmental protection and pursued high economic growth jointly with enterprises, resulting in serious water pollution problems. The behavior of local governments is closely related to their jurisdictions’ environmental conditions, and it is necessary to incorporate the behavior of local governments into the research framework for environmental issues.

Through strategic interactions, local governments fiercely compete in attracting investment and in realizing economic growth [21,22]. In order to attract high-quality projects, local governments compete to reduce environmental regulatory standards, increase the scale of industrial land transfers and the proportion of land supplied by agreement, and introduce bottom-line competition for low-quality investment. Due to the lack of an identification mechanism to distinguish good enterprises from bad ones, agreement transfers tend to absorb low-quality investment, such as enterprises with low-efficiency or high-pollution. There is competition between local governments to attract investment, which creates a “buyer’s market” for land. Therefore, local governments tend to engage in a “race to the bottom” in allocating industrial land and making concessions in land prices, labor costs, and environmental regulations. Local governments have a strong incentive to attract high-pollution enterprises, resulting in a “local regulation bias” [23]. Local governments that are geographically adjacent or that have similar levels of economic development will also compete to imitate each other’s land supply behavior. Fiscal incentives and land mismatches have resulted in a large supply of new construction land, which has led to excessive consumption of planned land quotas [24]. The central government promulgated the State Council’s Notice on Strengthening Land Control and Related Issues on 31 August 2006, which stipulated that, from 1 January 2007, industrial land must be sold by auction, listing, or open tender. However, the proportion of land transfers made by local government agreements remains relatively high, and the “race to the bottom” between local governments explains this phenomenon. In addition, the differentiated transfer strategies for different types of land are rational under financial incentives and external constraints [25], but they can easily lead to improper allocation of land resources, resulting in the misallocation of land resources in the industrial and service sectors [26]. Further, the misallocation of land resources has been found to have a negative impact on the allocation of financial lending [27], thereby adversely affecting both economic development and environmental protection. Most of the existing literature asserts that transfers of industrial land through agreements introduce relatively low-quality investment [28,29] and cause lower output efficiency [30,31], which results in more serious pollution. The method and pricing of local government industrial land supplies will affect the economic performance of regions and enterprises. Therefore, improving the method of allocating land resources can increase enterprise efficiency and promote regional economic and environmental development.

Although many scholars have broadly studied pollution and industrial land supply behavior by local governments, respectively, little research has focused on the impact of industrial land supply on pollution [32,33,34,35,36]. Due to data limitations and the heterogeneity of plot endowments, little existing literature discusses the rationales behind local governments’ industrial land supply behavior and its impact on environmental pollution in urban China. For clearer and more direct understanding on this paper, we provide a hypothesized conceptual model below (Figure 1) based on the above hypotheses and the whole structure of our paper.

## 3. Methods and Data

### 3.1. Research Methods

To investigate the impacts of local governments’ land supply behavior on environmental pollution, we constructed a data set comprising observations for 277 Chinese cities from 2009 to 2017. Our analysis includes the following steps (Figure 2):

Firstly, to provide a clear insight of the national level distribution and regional differences, we use GIS to display the spatial distribution of pollution (annual average concentration of PM2.5) and industrial land supply, respectively. By comparing the change of pollution with the ratio of industrial land supply, we could roughly deduce the qualitative impacts of land supply behavior on environmental pollution.

Secondly, to quantitatively evaluate the impacts of industrial land supply on environmental pollution, we use a two-way fixed effects regression model as our benchmark model. The two-way fixed effects model is now widely employed to estimate treatment effects. Chaisemartin and D’Haultfœuille [37] found that from 2010 to 2012, 19% of empirical articles in the *American Economic Review* (AER) applied the two-way fixed effects regression to evaluate the effect of a treatment on an outcome. It could adjust for two types of unobserved confounders (unit-specific and time-specific) at the same time [38] by including both sets of fixed terms, namely cities and years. The city fixed effects describe the permanent differences between cities. The year fixed effects capture the impacts that are common to all cities but vary by year. This two-way fixed effects model is a useful scientific and rigorous tool as it could provide more accurate econometric estimates and greatly alleviate the disturbance from multi-collinearity and thus improve the reliability of our conclusions. 

According to Allison [39], the general panel model has the following structure:(1)$$y_{i,t}=\sum_{j=0}^{J}\beta_{j}x_{i,t,j}+u_{it}$$

The one-way fixed effect model is a special case when we include one set of fixed effects in Equation (1) by assuming $u_{it}\;$= $\lambda_{i}+\varepsilon_{it}$:(2)$$y_{i,t}=\sum_{j=0}^{J}\beta_{j}x_{i,t,j}+\lambda_{i}+\varepsilon_{it}$$

Here, $\lambda_{i}$ is the unit-specific fixed effect, while ε*_it_* is the error term. To further include both the unit-specific and time-specific fixed effects, we could obtain a two-way fixed effects model by assuming *u_it_* = $\lambda_{i}+\gamma_{t}+\varepsilon_{it}$ in Equation (1):(3)$$y_{i,t}=\sum_{j=0}^{J}\beta_{j}x_{i,t,j}+\lambda_{i}+\gamma_{t}+\varepsilon_{it}$$

Now, $\gamma_{t}$ is the time-specific fixed effect. To investigate the effect of industrial land supply on the environment, we use a two-way fixed effects method for the benchmark model to conduct empirical analysis:(4)$$pollution_{i,t}=\alpha_{0}+\sum_{j=0}^{2}\beta_{j}induslandcov_{i,t-j}+\theta Control_{it}+\lambda_{i}+\gamma_{t}+\varepsilon_{it}$$

The subscripts *i* and *t* index the city and the year, respectively. *Pollution_i__,t_* is a set of variables measuring environmental pollution. $Induslandcov_{i,t-j}$ is the ratio of industrial and mining storage land to total land supply. *Control_it_* is a vector of control variables. *ε_it_* is the error term. $\lambda_{i}$ denotes city fixed effects and the controlling of time-invariant city-level effects. $\gamma_{t}$ denotes year fixed effects, capturing the aggregate trend in pollution and time-varying central policy shocks. The explained variable in this paper is the environmental pollution intensity index. Following the method of Song et al. [40], we use the following pollutants to measure pollution intensity: annual average concentration of PM2.5 (*lpm25*), annual average concentration of PM10 (*lpm10*), total carbon dioxide emissions (*lCO2*), and industrial soot and dust emissions (*ldust*). In benchmark regressions, we use the annual average concentration of PM2.5 as the core explanatory variable. Then, we use other pollutants mentioned above in robustness tests. The core explanatory variable is the industrial land supply. To attract investment and to promote industrial development, local governments often supply large amounts of industrial land through agreement. Therefore, we use the ratio of industrial and mining storage land to the total land supply (*induslandcov*) as the main explanatory variable to indicate the investment-attracting and land-transferring intensity of local governments. The main control variables include economic development, industrial structure, demographic patterns, international trade, and urban greening. We use GDP per square kilometer (*lggdp*), ratio of tertiary industry output value (*instr3gdp*), total output value of industrial enterprises above a designated size (*lindusvalue*), population density (*popdensity*), total actual foreign investment (*lFDI*), and green coverage rate of built-up (*green*) areas as measures of above factors. Variable definitions are provided in Table 1 below.

In order to examine the differences between regions, we further divide the sample into eastern, central, and western regions and perform group-level regressions. Thus, we can further investigate the regional heterogeneity in Section 5.1. 

Thirdly, we run the mechanism analysis from three dimensions: industrial structure, industrial development level, and local finance. Specifically, we first apply the classic two-step method to test the intermediary variables. Then, we run the Sobel test and the bootstrap test for the intermediary variables that do not meet the significant conditions of the two-step method. According to Table 1, we use the ratio of the secondary industry output value to the tertiary industry output value (*instr23*), the number of industrial enterprises above designated size (*lscalenumber*), and a fiscal balance index (*fiscalbalance*) as measures of the three possible mechanisms, respectively.

### 3.2. Data 

Industrial land supply data mainly come from the *Statistical Yearbook of Land and Resources* and city and environmental data mainly come from the *Statistical Yearbook of Environment*, the *Statistical Yearbook of Cities*, the *CSMAR* database, and the *China Economic Network Database*. We supplement some missing values using data from the *China Statistical Yearbook* and the *Regional Statistical Yearbook*. To avoid distortions from extreme values, we winsorize outliers at the 1st and 99th percentiles. Due to availability of information, our sample only includes data from mainland China.

Table 2 shows the summary statistics of the variables. Followed by the first column of variable name, the remaining five columns provide the number of observations, mean value, minimum value, maximum value, and standard deviation of each variable The core explanatory variable *induslandcov* has a mean value of 0.31 with an SD (standard deviation) of 0.16. The explained variable *lpm25* has a mean value of 3.71 with an SD of 047. For the rest of the pollution indicators, the mean values for *lpm10*, lCO2, lNOx, and ldust are 12.86, 3.15, 9.80, and 9.77, respectively. For the six main control variables: *lggdp* has a mean value of 6.95 with an SD of 1.28; the variable of the population density has a mean value of 0.04 with a standard deviation of 0.03; *lindusvalue* has a mean value of 16.63 and an SD of 1.20; *instr3gdp* has a mean of 37.77 with an SD of 9.12; *lFDI* has a mean value of 10.12 with an SD of 1.71; *green* has a mean value of 38.87 with an SD of 7.00. Similarly, the volatility of the three intermediary mechanism variables (*instr23*, *lscalenumber*, *fiscalbalance*) are measured as mean value and standard deviation (1.43 with an SD of 0.61; 6.58 with an SD of 1.10; 1.84 with an SD of 1.99). We could also observe a significant difference in the maximum and minimum values for each variable. For example, the maximum (0.84) of *induslandcov* is far greater than the minimum (0.00). Meanwhile, there exist substantial gaps in *lpm25* and *lpm10* (4.55 vs. 2.34 and 16.13 vs. 3.37), which reflect that the air pollution in cities have dispersed distributions.

## 4. Results

### 4.1. Spatial Distribution of PM2.5 and Industrial Land Supply

In 1986, the “Seventh Five-year Plan” divided mainland Chinese economic area into an east region, a middle region, and a west region. Here, we provide a figure (Figure 3) for readers to have a general understanding of China’s official division of regions.

The following Figure 4 provides a clear insight on the prefecture-level data of PM 2.5 during the years 2009 to 2017. According to the results, the spatial distribution of PM2.5 shows a high level of variation in each year from 2009 to 2017. The PM2.5 was higher in the central region and the eastern region than in the western region. Most cities experienced improving air quality, except for the northern part of the central region. As shown by Figure 4a, in the year 2009, we can easily observe a high level of PM 2.5 concentration in the central and eastern regions where most cities are colored in deep red (a deeper color means a higher concentration of PM 2.5). The northern region is mainly colored in lighter red, which means the air pollution there in 2009 was considerably less severe. In the southern and western regions, most cities suffered from a moderate level of pollution. Later, in the year 2011, the pollution became more serious in all three regions, as shown in Figure 4b,g,h. In the year 2013, however, some parts of the eastern region (Guangdong and Liaoning provinces) and the western region (Gansu province) experienced a reduction of PM2.5 concentration. As a comparison, in the year 2017 however, most cities experienced an improvement in PM2.5 concentration, where the colors become much lighter. The big “dark red zone” in the central and eastern regions contracted rapidly to a much smaller area (possibly in Hebei and Henan provinces only). In most parts of the southern and western regions, we can also observe a reduction in PM2.5 concentrations. However, some cities in the northeastern part of the central region (Jilin and Heilongjiang provinces) experienced a deterioration of air quality between 2009 and 2017. Figure 4f plots the change of PM2.5 concentration from 2009 to 2017. From this figure, we could confirm the observations above that most cities experienced a reduction of PM2.5 in 2017 compared with 2009. An obvious exception is the northern region, where most cities were exposed to a higher level of PM2.5 in 2017 instead. The reduction of air pollution is most obvious in the central and eastern regions, where many cities are colored in green.

In Figure 5, we provide a figure to illustrate the industrial land ratio from 2009 to 2017. According to the results, the spatial distribution of the industrial land ratio is quite uneven. The eastern region has the highest industrial land ratio, while the western region has the lowest. In all three regions, the industrial land ratio experienced a decreasing trend from 2009 to 2017. Coincidentally, some cities in the northern part of the central region have an increasing industrial land ratio. By observing Figure 5a,b, we know that, in 2009 and 2011, the industrial land ratio (*induslandcov*) was high in most of the cities. In some regions of the east coast (Guangdong, Jiangsu, Shandong, and Liaoning provinces), the central region (Heilongjiang and Hebei provinces), and the western part (Gansu province), the *induslandcov* even exceed 50%, which means that over half of the lands are supplied for industrial use. The relatively high *induslandcov* in the first few years may cause greater pollution in the following few years, which supports the increasing trend of PM2.5 concentration in Figure 4h from 2009 to 2011. From Figure 5c–f, we could easily conclude that most cities have a relatively moderate level of industrial land supply in 2013, 2015, and 2017. Compared with earlier years, *induslandcov* decreases significantly, which can be justified by observing Figure 5g,h. For the northeastern part of the central region (Heilongjiang and Jilin provinces), *induslandcov* remains at a high level during the whole period, therefore it is also the main region that experienced a deterioration of air quality in Figure 4f. 

According to the results from Figure 4 and Figure 5 above, we could observe a positive correlation between the PM2.5 concentration and the industrial land ratio. Regions with a higher industrial land ratio tend to also have a higher PM2.5 concentration. Therefore, we could qualitatively hypothesize that a higher industrial land ratio may be causing higher pollution. In order to portray the relationship between them more accurately and quantitatively, we would run a two-way fixed effects regression model in the next section and take different tests successively. 

### 4.2. Benchmark Regression Results

Previous studies show that most cities with high levels of environmental pollution are at an early stage of economic growth with relatively weak environmental regulations. Therefore, these cities are more likely to allocate construction land to the industrial sector that attracts high pollution enterprises. The time taken to construct an industrial building is generally 1–2 years, during which time the dust from a construction site may affect urban air quality indicators. To solve this problem, we use the proportion of industrial and mining storage land with lags of 1 and 2 years in different model specifications. The benchmark regression results are shown in Table 3. All regressions include both city and year fixed effects. In column (1), we introduce the core explanatory variable (*induslandcov*). The estimated coefficient is not statistically significantly different from zero, so the proportion of industrial land supply has no statistically significant impact on urban air quality in the same year. In columns (2) to (4), we introduce core explanatory variables with a one-year lag, two-year lag, and both lags, respectively.

The results show that the proportion of industrial land has a significant positive impact on regional environmental pollution only after two years and its impacts on environmental pollution in the current period and the next period are negative and insignificantly positive, respectively. A possible reason for this is that after the completion of a land transaction, it usually takes some time for enterprises to commence operating and we infer that it generally takes about two years to reach a normal level of production. In column (3), we find that the coefficient for the benchmark regression is 0.0623, significantly effective at the 1% test level. This number indicates that for a city with a PM2.5 concentration of 40μg/m^3^ in a particular year, a 10 percent increase in the industrial land ratio in that year alone will increase its PM2.5 concentration by 0.25μg/m^3^ after two years, leaving all other variables constant. The empirical results show that an increased supply of industrial land by the local government will increase regional environmental pollution. 

*lggdp*, *lindusvalue*, *popdensity*, *instr3gdp*, *lFDI*, and *green* are control variables that may also affect PM2.5 concentrations. For example, *lggdp* has a significant negative impact around 0.15 on an annual average PM2.5 concentration in all four columns, which infers that less developed areas are suffering more from air pollution possibly due to insufficient financial support for environmental protection. In addition, *lindusvalue* has a significantly negative impact on the annual average PM2.5 in the first three columns, indicating that the expansion of the industrial output value would decrease PM2.5 concentrations. The coefficient is decreasing in absolute value from column (1) to column (3), which means that the long-run impact decreases with time. Population density and green coverage rates have a positive but insignificant correlation with PM2.5 concentrations. The coefficient is increasing in absolute value from column (1) to column (3) for popdensity, which means that the long-run impact of population density increases with time. In column (3), we also find that *instr3gdp* has a positive impact on PM2.5 concentration, while *lGDP* has a negative impact. Our benchmark regression model has an adjusted R^2^ of 0.9502, which means that the core explanatory variable explains 95.02% of the variation in our dependent variable. 

### 4.3. Robustness Tests

In the robustness tests, we first select the annual average concentration of PM10, total carbon dioxide emissions, industrial nitrogen oxides emissions, and industrial soot dust emissions to replace the original explained variable. It shows that our benchmark regression results remain robust after replacing PM2.5 with alternative explained variables. We can find positive and significant coefficients in all columns of Table 4. To be more specific, the coefficients are effectively significant at the 1% test level in column (2), (3), and (4). For example, in column (3), we could find that a 10% increase in the ratio of industrial and mining storage land to total land supply would increase total NOx(tons) emission by 0.0265% after two years. We could conclude that, generally, the more industrial land is provided by local government, the worse the regional air quality, and the regression results remain robust if we use either one of *lpm10*, *lCO2*, *lNOx,* or *ldust* to replace PM2.5 concentration. 

Second, we narrow the sample range to further test the robustness of the benchmark results. It shows that our benchmark regression results remain robust after narrowing our samples. In the process of urban construction and development, province-level municipalities and central cities are more likely than other cities to receive special policy support. As such, the intensity of land control faced by their local governments will also differ from other cities. As a robustness test, we therefore exclude four municipalities (Beijing, Tianjin, Shanghai, and Chongqing) from our sample, and the results are reported in column (2) of Table 5. We find that the coefficient of *induslandcov(t−2)* becomes 0.0637 and remains statistically significant at the 1% level. Further, following Rao et al. [41], we also exclude provincial capital cities and sub-provincial cities from our sample and the results are reported in column (3) of Table 5. The empirical results show that the positive impact of industrial land expansion on environmental pollution remains robust for the small sample: the coefficient now is 0.0630 and the result is statistically significant at the 5% level. Compared with column (1), the regression coefficient on the core explanatory variable increases slightly from 0.0623 to 0.0630. This means that, compared with developed regions, local governments’ land supply in less-developed regions has greater externalities on regional environmental quality. Compared with the benchmark regression, *lggdp* still has a significant negative impact on PM2.5 concentration, the impact of lindusvalue now becomes insignificant, and all other control variables still have insignificant coefficients.

## 5. Heterogeneity and Mechanisms

### 5.1. Regional Heterogeneity Analysis

Table 6 shows that the positive impact of the proportion of industrial land on environmental pollution is statistically significant in the central and western regions. However, the impact becomes statistically insignificant and negative in the eastern region. More specifically, *induslandcov(t*−*2)* has positive coefficients (0.0876, and 0.0926) in the central and western regions, both significantly effective at the 5% level. Compared with the national average level of 0.0623, the coefficients in the central and western regions are also positive but become evidently higher, while the eastern region has a statistically insignificant negative coefficient of −0.0348. Due to the eastern region’s higher level of economic development, proportion of the service industry, degree of urbanization, and advancement of industrial transformation and upgrading, in this region the impact of industrial enterprises on environmental pollution is relatively small. In contrast, due to the abundant land area in inland China, the human–land contradiction is relatively moderate in the central and western regions. In addition, the mode of economic development and types of industry are relatively homogeneous in the central and western regions. Further, the central and western regions have abundant construction land quotas, so local governments are more inclined to adopt differentiated land supply strategies to promote regional economic development through industrialization. Therefore, compared with the eastern region, the land transfer behavior of local governments in the central and western regions has a greater impact on environmental pollution. To solve the current problems of high population and limited land in the eastern region and low population and abundant land in the central and western regions, many scholars have argued that the allocation of construction land quotas should closely follow population flows. In addition, the regional heterogeneity in the impact of land transfer behavior on environmental pollution has certain inspiring significance for the formulation of land transfer regulatory policies in different regions.

### 5.2. Mechanism Research

In this part, we discuss the mechanism linking the availability of industrial land and pollution from three dimensions: industrial structure, industrial development level, and local finance. Table 7 and Table 8 report the results of the two-step method, the Sobel test, and the bootstrap test. 

According to the results, the industrial structure does not constitute an intermediary mechanism. The misallocation of land resources adversely affects environmental quality by hindering the upgrading of local industrial structures [42,43]. In this paper, the proportion of the output value of secondary industry and the output value of tertiary industry are used to measure the degree of equilibrium of the industrial structure. Regression results for the intermediary effect are shown in columns (1)–(2) of Table 7. In column (2), empirical results show that the coefficient on *instr23* is not statistically significant even at the 10% level. Therefore, the industrial structure does not constitute an intermediary mechanism for the relationship between regional land supply behavior and environmental quality. Nevertheless, when local governments increase the supply of industrial land, this will increase the proportion of local secondary industry and the possible imbalance in the industrial structure is still worthy of attention.

According to Table 7 and Table 8, the scale expansion effect constitutes an intermediary mechanism. Industrial enterprises above a designated size have a large production scale and high-tech equipment. It is easier for such enterprises to rely on technological improvements to achieve reasonable emissions and green production. Their production behavior will cause less environmental damage than other enterprises to the local environment and may even be beneficial to environmental governance. We use the number of industrial enterprises above a designated size as a proxy variable for the level of regional industrial development. Based on the results of the two-step method, an increase in the proportion of industrial land used by local governments positively affects the number of industrial enterprises above the designated size in the region, but the coefficient is only 0.0018 and is statistically insignificant. Further, an increase in the number of industrial enterprises above the designated size is associated with a limited improvement in environmental pollution. The coefficient is insignificantly negative at −0.0148. This shows that the local governments’ expectation of promoting regional industrial development through the “land-for-investment” method often fails to bring about high-quality industrial development. Instead, the entry of other industrial enterprises may crowd out development resources of the original enterprises in the region whose production has reached the designated size. Furthermore, because some newly settled enterprises cannot meet strict pollution discharge standards and environmental regulations, their production behaviors are detrimental to the environment in the region. However, due to the results from the Sobel test and the bootstrap test, the scale expansion effect now has a significant intermediary effect.

According to Table 7 and Table 8, fiscal imbalance constitutes an intermediary mechanism. As an important source of local fiscal revenue, land transfer fees support the construction and the governance of cities. Following the previous literature, we use revenue and expenditure reported in local fiscal budgets to measure the regional balance of payments and include this variable in our regressions. The results are reported in columns (5)–(6) of Table 7. In column (6), the coefficients for *fiscalbalance* and *induslandcov(t−2)* are 0.0106 and 0.0635, which are statistically significant at the 1% and the 5% level, respectively. These results show that greater industrial land transfers by a local government lead to an increase in the local government’s fiscal deficit and this increase in fiscal deficit is associated with increased pollution. On the one hand, compared with the transfer methods of bidding, auction, and listing, the amount of local fiscal revenue from agreement transfer fees is very small. On the other hand, a larger number of land transactions will require local governments to increase financial expenditures. However, these expenditures are used more for urban construction than for environmental governance so may not effectively contribute to mitigating pollution.

## 6. Conclusions

Using the data of 277 prefecture-level cities in China from 2009 to 2017, this study quantitatively investigates the impact of industrial land supply by local governments on regional environmental pollution. The findings suggest that when local governments increase the proportion of industrial land, they will aggravate regional environmental pollution; this supports the qualitative inferences from our spatial distribution analysis. The coefficient is 0.0623 and is statistically significant at the 1% level. The positive impact of the proportion of industrial land on environmental pollution is stronger in the central and western regions; the coefficients on explanatory variables in the central and western regions are 0.0876 and 0.0926, respectively, and are significantly higher than the national average of 0.0623. Meanwhile, the eastern region has a statistically insignificant negative coefficient of −0.0348. Further, the results remain robust and significant after using alternative explained variables or a narrowing sample range. Lastly, we find that the expansion scale effect and imbalanced finance form two valid and significant intermediary mechanisms.

Similar to western countries [44,45] and other third world countries [46], our article finds a positive impact of industrial land supply on environmental pollution. Some Chinese scholars studied the land misallocation on pollution [47], air quality [48], and its spatial spillover effects [49]. Zhao and Yan [50] studied the change of industrial land supply on environment in Chinese silk road cities. They usually use the spatial Durbin model and mainly focus on the consequences of regional mismatches. However, they fail to explain the possible reasons for such a mismatch. Our article contributes the existing literature in the following aspects. First, based on industrial land supply data for prefecture-level cities from the *National Land and Resources Yearbook*, this paper describes the behavior of local governments in industrial land selling more accurately and comprehensively than prior studies and explains regional environmental pollution from this perspective. Second, this paper considers the effects of government officials’ promotion incentives and interactions between local governments. We explore the environmental externalities brought about by differences in local governments’ land transfer behaviors from the perspective of land resource allocation and provide empirical evidence for how the institutional design of regional land resource allocation can be improved. Thirdly, we apply a two-way fixed effects model in our benchmark regression that is rigorous and scientific. Finally, this paper analyzes the mechanisms behind the environmental externalities caused by local governments’ land allocation behavior. 

We acknowledge the limitations of our study. First, we use a two-way fixed effects model in our article to simultaneously adjust for cities’ and years’ fixed effects. However, whether these two types of unobserved confounders can be adjusted at the same time relies on the assumption of linear additive effects [38], which may need further examination. In addition, we only investigate the impact of industrial land supply and aggregate all other land use types as “others”. Future researchers could do more analysis on other different land use types, such as residential and commercial land, respectively, to explore whether these types of land supply have certain impacts on environmental pollution. In addition, the formation and spread of environmental pollution is not limited to the city in the administrative sense; there are often spillover effects of environmental pollution between regions. Whether land transfer competition between regions will deepen mismatches of land resources and thus aggravate environmental pollution, and whether there are other mechanisms by which local government land transfer affects environmental pollution in the jurisdiction area, are topics that deserve further study.

Based on the above findings, this paper proposes the following policy recommendations:(1)Local governments should adhere to the ecological bottom line, adjust measures to local conditions, and formulate land transfer regulations and environmental protection policies at a regional level. According to the specific conditions of different regions, it is necessary to allow certain differences and particularities in policy formulation and to formulate land transfer regulations and environmental protection policies in a targeted manner to avoid a “one size fits all” approach. For the eastern and central regions, the use and allocation of construction land indicators by local governments should be regulated by strengthening legislation and administrative supervision to alleviate problems such as price distortions and structural distortions in different types of land transfers. The national government should actively promote the secondary land market transactions and the circulation of land and optimize the allocation of resources. For the central and western regions, land resource assessment and planning should be performed more scientifically to improve the efficiency of land resource utilization.(2)Local governments should balance the concepts of attracting investment and ensuring green development. Local governments should abandon development concepts such as “GDP only” and “GDP first” and plan development from a longer-term perspective. In the process of attracting investment, local governments should establish an identification mechanism to eliminate some enterprises with low production efficiency and high pollution emissions. At the same time, they could encourage green production by existing enterprises using financial subsidies and other means.(3)Local governments should pay more attention to environmental governance and should especially allocate money for this purpose. Local governments should standardize land market transactions, rationally plan the structure of fiscal revenue and expenditure, and set up special funds for environmental governance. They should also accelerate the progress of pollution control and the construction of an environmental protection infrastructure within their jurisdictions and should provide funding for the development of the ecological environment in their regions.

## Figures and Tables

**Figure 1 ijerph-19-14890-f001:**
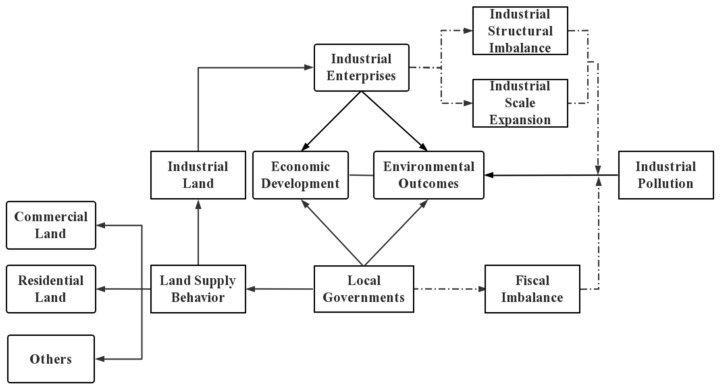
Hypothesized conceptual framework for understanding the impacts of industrial land supply on environmental pollution.

**Figure 2 ijerph-19-14890-f002:**
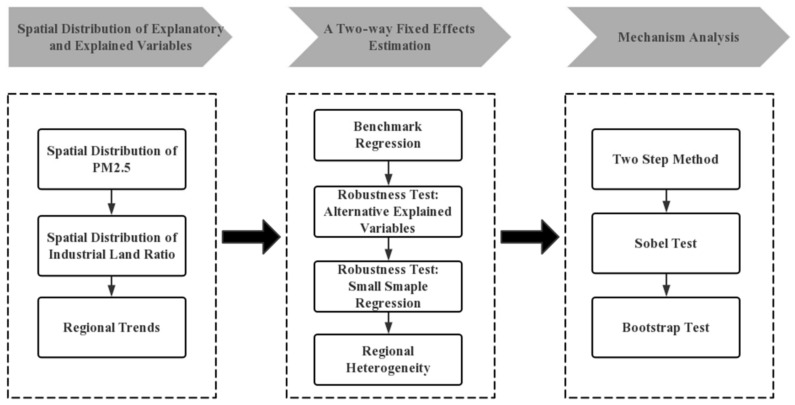
A brief methodology framework.

**Figure 3 ijerph-19-14890-f003:**
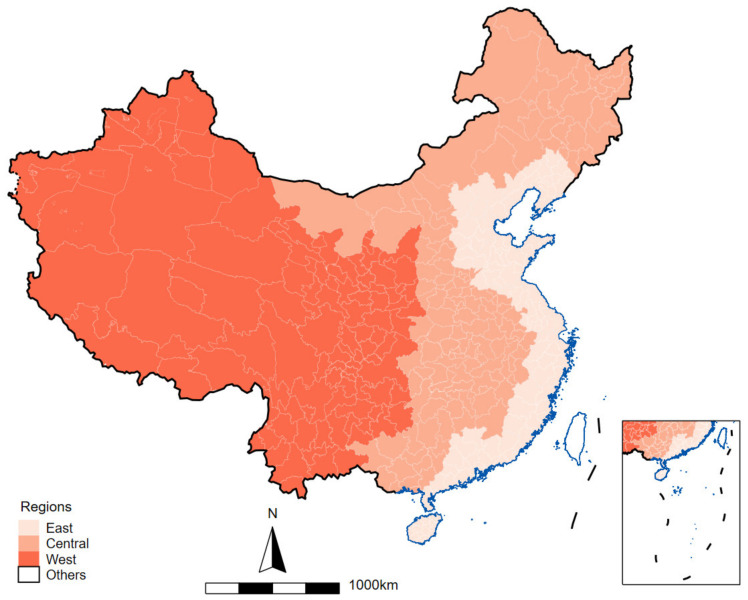
Division of Western, Central, and Eastern China.

**Figure 4 ijerph-19-14890-f004:**
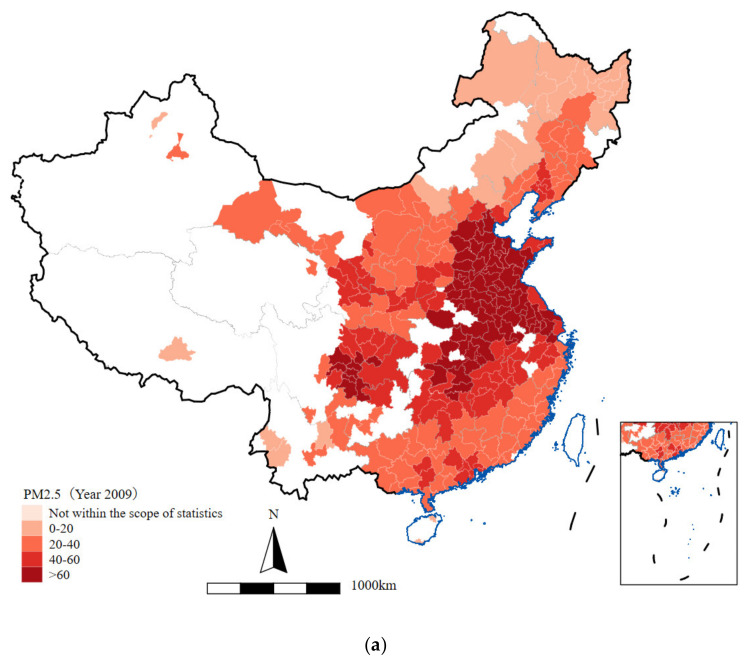
(**a**–**e**) PM2.5 in 2009, 2011, 2013, 2015, and 2017; (**f**) PM2.5 change from 2009 to year 2017; (**g**,**h**) regional PM2.5 concentration from 2009 to 2017.

**Figure 5 ijerph-19-14890-f005:**
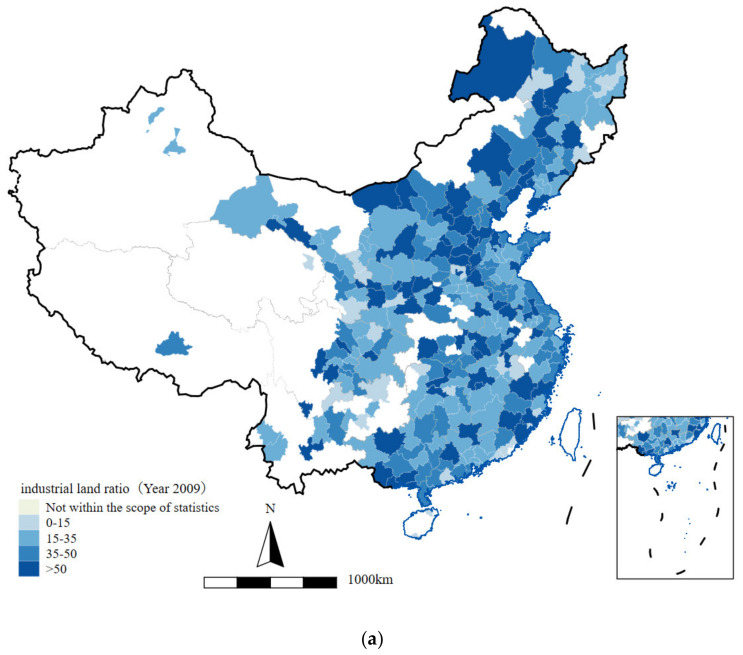
(**a**–**f**) Industrial land ratio in 2009, 2011, 2013, 2015, 2016, and 2017; (**g**,**h**) regional industrial land ratio from 2009 to 2017.

**Table 1 ijerph-19-14890-t001:** Variable definitions.

Variable Name	Definition
*lpm25*	Annual average concentration of PM2.5 (μg/m^3^)
*lpm10*	Annual average concentration of PM10 (μg/m^3^)
*lCO_2_*	Total carbon dioxide emissions (million tons)
*lSO_2_*	Industrial sulfur dioxide emissions (tons)
*lNOx*	Industrial nitrogen oxide emissions (tons)
*ldust*	Industrial soot and dust emissions (tons)
*lwastewater*	Industrial waste water emissions (ten thousand tons)
*indusland*	Supply of industrial and mining storage land (hectares)
*induslandcov*	Ratio of industrial and mining storage land to total land supply (%)
*lggdp*	GDP per square kilometer (hundred million RMB per square kilometer)
*lindusvalue*	Total output value of industrial enterprises above designated size (ten thousand RMB)
*popdensity*	Population density (people per square kilometer)
*instr3gdp*	Share of tertiary industry (%)
*instr23*	Ratio of secondary industry output value to tertiary industry output value
*lFDI*	Total actual foreign investment (USD 10,000)
*green*	Green coverage rate of built-up area (%)
*lscalenumber*	Number of industrial enterprises above designated size
*fiscalbalance*	(Expenditures within the local budget − revenue within the budget) ÷ revenue within the budget

**Table 2 ijerph-19-14890-t002:** Summary statistics for key variables.

Variable	Observations	Mean Value	Minimum	Maximum	Standard Deviation
*lpm25*	2517	3.71	2.34	4.55	0.47
*lpm10*	2365	12.86	3.37	16.13	3.28
*lCO_2_*	2511	3.15	1.72	4.87	0.71
*lNOx*	2397	9.80	6.72	12.22	1.04
*ldust*	2398	9.77	6.72	12.23	1.07
*indusland*	2524	489.44	10.81	2688.78	474.64
*induslandcov*	2524	0.31	0.00	0.84	0.16
*lggdp*	2516	6.95	3.83	10.08	1.28
*lindusvalue*	2340	16.63	13.30	19.30	1.20
*popdensity*	2520	0.04	0.00	0.20	0.03
*instr3gdp*	2235	37.77	18.81	67.76	9.12
*instr23*	2234	1.43	0.31	3.97	0.61
*lFDI*	2314	10.12	5.30	13.89	1.71
*green*	2471	38.87	10.10	57.71	7.00
*lscalenumber*	2522	6.58	3.85	9.00	1.10
*fiscalbalance*	2522	1.84	−0.35	26.58	1.99

**Table 3 ijerph-19-14890-t003:** Benchmark regression results.

Variables	(1)	(2)	(3)	(4)
*induslandcov*	−0.0092			
	(−0.4453)			
*induslandcov(t*−*1)*		0.0090		0.0219
		(0.4086)		(0.8680)
*induslandcov(t*−*2)*			0.0623 ***	0.0638 ***
			(2.6063)	(2.6683)
*lggdp*	−0.1492 ***	−0.1710 ***	−0.1486 ***	−0.1351 ***
	(−5.1650)	(−5.4732)	(−4.3957)	(−4.0031)
*lindusvalue*	−0.0354 ***	−0.0257 *	−0.0252 *	−0.0212
	(−2.6985)	(−1.8567)	(−1.6849)	(−1.4201)
*popdensity*	0.0202	0.1226	0.2362	0.2182
	(0.0918)	(0.5668)	(1.0344)	(0.9628)
*instr3gdp*	−0.0011	0.0010	0.0015	0.0016
	(−0.8139)	(0.6585)	(0.8752)	(0.9416)
*lFDI*	0.0006	−0.0039	−0.0026	−0.0010
	(0.1399)	(−0.8807)	(−0.5409)	(−0.2022)
*green*	0.0006	0.0005	0.0004	0.0006
	(1.0225)	(0.7779)	(0.5029)	(0.8435)
*Year fixed effects*	YES	YES	YES	YES
*City fixed effects*	YES	YES	YES	YES
Observations	2077	1783	1519	1491
Adjusted R^2^	0.9476	0.9501	0.9502	0.9500

Note: the dependent variable is *lpm25*; values in parentheses are t-values; * and *** denote statistical significance at the 10% and 1% levels, respectively.

**Table 4 ijerph-19-14890-t004:** Robustness test.

Variables	(1)	(2)	(4)	(5)
	*lpm10*	*lCO_2_*	*lNOx*	*ldust*
*induslandcov(t*−*2)*	0.2795 *	0.0277 ***	0.2648 ***	0.2649 ***
	(1.6665)	(2.9808)	(2.6639)	(2.6646)
*lggdp*	0.1060	0.0112	0.0668	0.0669
	(0.4407)	(0.8509)	(0.4664)	(0.4669)
*lindusvalue*	0.0453	0.0060	0.0915	0.0915
	(0.3358)	(1.0341)	(1.1466)	(1.1471)
*popdensity*	−3.3910 **	0.1187	−0.5951	−0.5955
	(−2.1515)	(1.3377)	(−0.6388)	(−0.6392)
*instr3gdp*	−0.0185	−0.0014 **	−0.0069	−0.0069
	(−1.4478)	(−2.0763)	(−0.9253)	(−0.9252)
*lFDI*	−0.0021	−0.0016	−0.0379 *	−0.0379 *
	(−0.0598)	(−0.8507)	(−1.8489)	(−1.8495)
*green*	0.0070	−0.0003	−0.0014	−0.0014
	(1.4005)	(−1.0249)	(−0.4775)	(−0.4774)
*Year fixed effects*	YES	YES	YES	YES
*City fixed effects*	YES	YES	YES	YES
Observations	1462	1519	1451	1451
Adjusted R^2^	0.7712	0.9967	0.8348	0.8348

Note: values in parentheses are t-values; *, **, and *** denote statistical significance at the 10%, 5%, and 1% levels, respectively.

**Table 5 ijerph-19-14890-t005:** Small sample regression.

Variables	(1)	(2)	(3)
*induslandcov(t*−*2)*	0.0623 ***	0.0637 ***	0.0630 **
	(2.6063)	(2.6574)	(2.5001)
*lggdp*	−0.1486 ***	−0.1459 ***	−0.1443 ***
	(−4.3957)	(−4.2879)	(−4.0731)
*lindusvalue*	−0.0252 *	−0.0226	−0.0206
	(−1.6849)	(−1.5026)	(−1.3330)
*popdensity*	0.2362	0.2279	0.2405
	(1.0344)	(0.9963)	(1.0057)
*instr3gdp*	0.0015	0.0021	0.0024
	(0.8752)	(1.1839)	(1.2649)
*lFDI*	−0.0026	−0.0027	−0.0020
	(−0.5409)	(−0.5528)	(−0.3910)
*green*	0.0004	0.0003	0.0003
	(0.5029)	(0.3970)	(0.4749)
*Year fixed effects*	YES	YES	YES
*City fixed effects*	YES	YES	YES
Observations	1519	1500	1353
Adjusted R^2^	0.9502	0.9503	0.9502

Note: the dependent variable is *lpm25*; values in parentheses are t-values; *, **, and *** denote statistical significance at the 10%, 5%, and 1% levels, respectively.

**Table 6 ijerph-19-14890-t006:** Regional heterogeneity analysis.

Variables	Eastern Region	Central Region	Western Region
*induslandcov(t*−*2)*	−0.0348	0.0876 **	0.0926 **
	(−1.1315)	(2.0545)	(2.0777)
*Control variables*	YES	YES	YES
*Year fixed effects*	YES	YES	YES
*City fixed effects*	YES	YES	YES
Observations	572	609	250
Adjusted R^2^	0.9700	0.9502	0.9643

Note: the dependent variable is *lpm25*; values in parentheses are t-values; ** denotes statistical significance at the 5%.

**Table 7 ijerph-19-14890-t007:** Mechanism analysis.

Variables	(1)	(2)	(3)	(4)	(5)	(6)
	Dependent Variable					
	*instr23*	*lpm25*	*lscalenumber*	*lpm25*	*fiscalbalance*	*lpm25*
*induslandcov(t*−*2)*	0.0652 ***	0.0608 **	0.0018	0.0623 ***	−0.0693	0.0635 **
	(2.8849)	(2.5328)	(0.0585)	(2.6067)	(−0.3905)	(2.4675)
*instr23*		0.0195				
		(0.6447)				
*lscalenumber*				−0.0148		
				(−0.6553)		
*fiscalbalance*						0.0106 ***
						(3.1679)
*Control variables*	YES	YES	YES	YES	YES	YES
*Year fixed effects*	YES	YES	YES	YES	YES	YES
*City fixed effects*	YES	YES	YES	YES	YES	YES
Observations	1518	1518	1519	1519	1518	1518
Adjusted R^2^	0.9715	0.9502	0.9835	0.9502	0.7445	0.9505

Note: values in parentheses are t-values; ** and *** denote statistical significance at the 5% and 1% levels, respectively; control variables include: *lggdp*, *lindusvalue*, *popdensity*, *instr3gdp*, and *green*.

**Table 8 ijerph-19-14890-t008:** Intermediary variable: Sobel and bootstrap test results.

Indicators	Equilibrium Degree of Industrial Structure	Industrial Enterprise Scale	Local Finance
	*Instr23*	*lscalenumber*	*fiscalbalance*
Sobel test coefficient	0.007	0.062 ***	−0.024 ***
* p* value	0.263	<0.001	0.002
Bootstrap test coefficient	0.007	0.062 ***	−0.024 ***
* p* value	0.29	<0.001	0.001

Note: *, **, and *** denote statistical significance at the 10%, 5%, and 1% levels, respectively.

## Data Availability

Some or all data and models that support the findings of this study are available from the corresponding author upon reasonable request.

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
