# Peer review of "A Two-Way Fixed Effects Estimation on the Impact of Industrial Land Supply on Environmental Pollution in Urban China"

_ijerph, 2022, doi:10.3390/ijerph192214890_

Round 1

Reviewer 1 Report (New Reviewer)

1. Shorten the Introduction section.

2. The conclusion is also very lengthy. Shorten it.

Author Response

Response to Reviewer 1 Comments

Dear Reviewer,

Thank you for your helpful suggestions and comments. We have revised our manuscript during the past week. Here are the main changes to be noticed:

Point 1: Shorten the Introduction section.

Response 1: The introduction section first introduces the research background. Then it states the basis, aim and scope of the research. Lastly it provides a brief organization of the following sections. The second paragraph contains some redundant statement, so we deleted some part of it:

Existing research has recognized the important role of the local governments' land resource allocation behavior on environmental problems. In the context of fiscal decentralization and the "GDP-growth tournament among governments", local government officials are generally inclined to adopt the model of "developing through land" due to financial and political career promotion incentives [2]. They tend to pursue regional economic growth goals and maximize fiscal revenue without fully considering regional environmental protection and governance. Specifically, local governments tend to adopt the method of transferring industrial land by agreement and apply a low entry threshold for absorbing investment to attract industrial enterprises to the region and  relax regional environmental regulations which aggravates environmental pollution. In addition, the extensive development model of "land for capital," selling industrial land cheaply, can quickly increase regional fixed asset investment, cultivate new tax sources, make up the gap between fiscal revenue and expenditure, and improve economic growth. With the advantage of a "two-way monopoly" in land transactions supported by China's land property rights system [3], local governments expropriate land from rural areas at a price lower than the land's value, and then supply land for real estate, commerce, and service industries at a high price to gain considerable land transfer fees. This differentiated land allocation strategy for industrial, commercial, and residential land is local governments' optimal strategic choice to maximize their own utility. However, the resulting inefficient supply of industrial land and damage to the ecological environment needs to be carefully evaluated, but it has not been to date.

Point 2: The conclusion is also very lengthy. Shorten it.

Response 2: We improve our conclusion by re-organize it to Conclusions and Discussions:

  1. Conclusions and Discussions

Using data of 277 prefecture-level cities in China from 2009 to 2017, this study quantitatively investigates the impact of industrial land supply by local governments on regional environmental pollution. The findings suggest that when local governments increase the proportion of industrial land, they will aggravate regional environmental pollution which supports the qualitative inferences from our spatial distribution analysis. The coefficient is 0.0623 and is statistically significant at 1% level. The positive impact of the proportion of industrial land on environmental pollution is stronger in the central and western regions, and the coefficients on explanatory variables in the central and western regions are 0.0876 and 0.0926 respectively and are significantly higher than the national average of 0.0623. Meanwhile, the eastern region has a statistically insignificant negative coefficient of -0.0348. Further, the results remain robust and significant after using alternative explained variables or narrowing sample range. Lastly, we find that expansion scale effect and imbalanced finance form two valid and significant intermediary mechanisms.

Similar to western countries [44-45] and other third world countries [46], our article find a positive impact of industrial land supply on environmental pollution. Some Chinese scholars studied the land misallocation on pollution [47], air quality [48] and its spatial spillover effects [49]. They usually uses spatial Durbin model and mainly focus on the consequences of regional mismatches. However, they fail to explain the possible reasons for such a mismatch. Our article contributes the existing literature in the following aspects. First, based on industrial land supply data for prefecture-level cities from the National Land and Resources Yearbook, this paper describes the behavior of local governments in industrial land selling more accurately and comprehensively than prior studies, and explains regional environmental pollution from this perspective. Second, this paper considers the effects of government officials' promotion incentives and interactions between local governments. We explore the environmental externalities brought about by differences in local governments' land transfer behaviors from the perspective of land resource allocation and provide empirical evidence for how the institutional design of regional land resource allocation can be improved. Thirdly, we apply a two-way fixed effects model in our benchmark regression which is rigorous and scientific. Finally, this paper analyzes the mechanisms behind the environmental externalities caused by local governments' land allocation behavior.

We acknowledge the limitations of our study. First, we use a two-way fixed effects model in our article to simultaneously adjust for cities and years fixed effects. However, whether these two types of unobserved confounders can be adjusted at the same time relies on the assumption of linear additive effects [38], which may need further examination. In addition, we only investigate the impact of industrial land supply and aggregate all other land use types as “others”. Future researchers could do more analysis on other different land use types, such as residential and commercial land respectively, to explore if these types of land supply have certain impacts on environmental pollution. In addition, the formation and spread of environmental pollution is not limited to the city in the administrative sense, there are often spillover effects of environmental pollution between regions. Whether land transfer competition between regions will deepen mismatches of land resources and thus aggravate environmental pollution, and whether there are other mechanisms by which local government land transfer affects environmental pollution in the jurisdiction area, are topics that deserve further study.

Based on the above findings, this paper proposes the following policy recommendations:

(1) Local governments should adhere to the ecological bottom line, adjust measures to local conditions, and formulate land transfer regulations and environmental protection policies at a regional level. According to the specific conditions of different regions, it is necessary to allow certain differences and particularities in policy formulation, and to formulate land transfer regulations and environmental protection policies in a targeted manner to avoid a "one size fits all" approach. For the eastern and central regions, the use and allocation of construction land indicators by local governments should be regulated by strengthening legislation and administrative supervision, to alleviate problems such as price distortions and structural distortions in different types of land transfers. The national government should actively promote the secondary land market transactions and circulation of land, and optimize the allocation of resources. For the central and western region, land resource assessment and planning should be performed more scientifically to improve the efficiency of land resource utilization.

(2) Local governments should balance the concepts of attracting investment and ensuring green development. Local governments should abandon development concepts such as "GDP only" and "GDP first," and plan development from a longer-term perspective. In the process of attracting investment, local governments should establish an identification mechanism to eliminate some enterprises with low production efficiency and high pollution emissions. At the same time, they could encourage green production by existing enterprises using financial subsidies and other means.

(3) Local governments should pay more attention to environmental governance, and should especially allocate money for this purpose. Local governments should standardize land market transactions, rationally plan the structure of fiscal revenue and expenditure, and set up special funds for environmental governance. They should also accelerate the progress of pollution control and construction of environmental protection infrastructure within their jurisdictions, and should provide funding for development of the ecological environment in their regions.

Other Changes:

  1. In the original version, our aims are not clearly mentioned in the introduction. We did not management to give a brief structure and relevant aim of each part of our research design so we improve this part:

To fill this gap, this study aims to investigate the impact of industrial land supply on environmental pollution in China through the combination of spatial analysis and regression models. We first use GIS to investigate the spatial variation of pollution and industrial land supply. We then apply a two-way fixed effects regression method to quantitatively evaluate the impacts of industrial land supply on environmental pollution. We further investigate the regional heterogeneity rigorously through group-level regressions to see whether the differentiated land allocation strategies adopted by local governments can aggravate regional environmental pollution. To test the robustness, we use alternative explained variables and narrow the samples successively. Further, we explore the intermediate mechanisms from three dimensions through two-step method, the Sobel test and bootstrap test. Lastly, we provide policy implications for optimizing the allocation of land resources, and improving land transfer regulations and environmental protection policies.

  1. We include a new Figure to indicate the West - Middle - East regions in section.

4.1. Spatial distribution of PM2.5 and industrial land supply

In 1986, "seven five" plan divided mainland Chinese economic area into east-region, middle-region and west-region. Here, we provide a figure for readers to have a general understanding of Chinas official division of regions.

Figure 3. Division of Western, Central and Eastern China

  1. In order to emphasize the main method that we use in this article, we change the title from A Quantitative Estimation on the Impact of Industrial Land Supply on Environmental Pollution in Urban China from 2009 to 2017to A Two-way Fixed Effect Estimation on the Impact of Industrial Land Supply on Environmental Pollution in Urban China.

There are also some other revisions and grammar corrections in our revised submission and all changes can be traced and viewed in our new MS Word Template.

Yours sincerely

The Authors

Reviewer 2 Report (New Reviewer)

Comments of Reviewer: Manuscript ID ijerph-1998036

Entitled „A Quantitative Estimation on the Impact of Industrial Land Supply on Environmental Pollution in Urban China from 2009 to 2017” by Xiangqi et al.

The introduction of the manuscript is well written, gives us a new and interesting topic, a new way how to communicate and advise decision-makers with knowledge on environmental issues about land use, pollution, etc. The introduction part covers the most significant existing literature. The used statistical methods are tolerably appropriate for the measurements. The results are clear and concise. As most cases, the first half of the manuscript deals with a precise, prudent use of literature background, but after collecting the information, the authors start to use a lot of different, sometimes controversial statistical analysis. At the end, the reader forgets and get lost of traction.

General comments:

1.      line 39: qualified what? missing information in the sentence.

2.      line40: those are not problems on sustainable development, they are the cause of it. Please clarify it.

3.      line41-44: these two sentences are controversial to each other, please clarify.

4.      line49: and what are that research?

5.      line100-101: Canada belongs to North America, and Spain not a region (write South-West Europe)

6.      Figure 1: Why is the legend in bold?

7.      Table 2: please indicate the meaning of the abbreviations

8.      line300-317: it is a result – not the data source -, please move this part into the result section

9.      Figure 3: too small, I have got good eyes, but I cannot read the years included on the separate figures. (a) is 2011? and (b) is 2009? if I can read it well, why these are not in order?

10.  Figure 4: subfigures are not in the correct place, same stands here- too small.

Specific comments:

·         Introduction and Literature review: I do not feel that these need to be a separate chapter. Please cover both in just the Introduction part.

·         My first main concern is that what are the aims? They are not mentioned clearly in the introduction.

·         In the Discussion chapter (which is missing), there are scarce amount of cited literature, I do not feel that all the written information stands alone without any prove/comparison!

·         What parts of China considered to be the given 3 regions? Please indicate the exact borders!

I know that the manuscript had a huge work in it, but the authors faced the biggest mistake that want to show more that they had.

Author Response

Dear Reviewer,

Thank you for your helpful suggestions and comments. We have revised our manuscript during the past week. Here are the main changes to be noticed:

Point 1:  My first main concern is that what are the aims? They are not mentioned clearly in the introduction.

Response 1: In the original version, our aims are not clearly mentioned in the introduction. We did not management to give a brief structure and relevant aim of each part of our research design so we improve this part:

To fill this gap, this study aims to investigate the impact of industrial land supply on environmental pollution in China through the combination of spatial analysis and regression models. We first use GIS to investigate the spatial variation of pollution and industrial land supply. We then apply a two-way fixed effects regression method to quantitatively evaluate the impacts of industrial land supply on environmental pollution. We further investigate the regional heterogeneity rigorously through group-level regressions to see whether the differentiated land allocation strategies adopted by local governments can aggravate regional environmental pollution. To test the robustness, we use alternative explained variables and narrow the samples successively. Further, we explore the intermediate mechanisms from three dimensions through two-step method, the Sobel test and bootstrap test. Lastly, we provide policy implications for optimizing the allocation of land resources, and improving land transfer regulations and environmental protection policies.

Point 2:  In the Discussion chapter (which is missing), there are scarce amount of cited literature, I do not feel that all the written information stands alone without any prove/comparison!

Response 2: Thank you for your advice here. We decided to include a Discussion section and combine it with the Conclusion. We Here is the Conclusions and Discussions:

  1. Conclusions and Discussions

Using data of 277 prefecture-level cities in China from 2009 to 2017, this study quantitatively investigates the impact of industrial land supply by local governments on regional environmental pollution. The findings suggest that when local governments increase the proportion of industrial land, they will aggravate regional environmental pollution which supports the qualitative inferences from our spatial distribution analysis. The coefficient is 0.0623 and is statistically significant at 1% level. The positive impact of the proportion of industrial land on environmental pollution is stronger in the central and western regions, and the coefficients on explanatory variables in the central and western regions are 0.0876 and 0.0926 respectively and are significantly higher than the national average of 0.0623. Meanwhile, the eastern region has a statistically insignificant negative coefficient of -0.0348. Further, the results remain robust and significant after using alternative explained variables or narrowing sample range. Lastly, we find that expansion scale effect and imbalanced finance form two valid and significant intermediary mechanisms.

Similar to western countries [44-45] and other third world countries [46], our article find a positive impact of industrial land supply on environmental pollution. Some Chinese scholars studied the land misallocation on pollution [47], air quality [48] and its spatial spillover effects [49]. They usually uses spatial Durbin model and mainly focus on the consequences of regional mismatches. However, they fail to explain the possible reasons for such a mismatch. Our article contributes the existing literature in the following aspects. First, based on industrial land supply data for prefecture-level cities from the National Land and Resources Yearbook, this paper describes the behavior of local governments in industrial land selling more accurately and comprehensively than prior studies, and explains regional environmental pollution from this perspective. Second, this paper considers the effects of government officials' promotion incentives and interactions between local governments. We explore the environmental externalities brought about by differences in local governments' land transfer behaviors from the perspective of land resource allocation and provide empirical evidence for how the institutional design of regional land resource allocation can be improved. Thirdly, we apply a two-way fixed effects model in our benchmark regression which is rigorous and scientific. Finally, this paper analyzes the mechanisms behind the environmental externalities caused by local governments' land allocation behavior.

We acknowledge the limitations of our study. First, we use a two-way fixed effects model in our article to simultaneously adjust for cities and years fixed effects. However, whether these two types of unobserved confounders can be adjusted at the same time relies on the assumption of linear additive effects [38], which may need further examination. In addition, we only investigate the impact of industrial land supply and aggregate all other land use types as “others”. Future researchers could do more analysis on other different land use types, such as residential and commercial land respectively, to explore if these types of land supply have certain impacts on environmental pollution. In addition, the formation and spread of environmental pollution is not limited to the city in the administrative sense, there are often spillover effects of environmental pollution between regions. Whether land transfer competition between regions will deepen mismatches of land resources and thus aggravate environmental pollution, and whether there are other mechanisms by which local government land transfer affects environmental pollution in the jurisdiction area, are topics that deserve further study.

Point 3: (line 39) qualified what? missing information in the sentence.

Response 3: qualified for drinking use.

83.4% of the national surface water sections are qualified. to 83.4% of the national surface water sections satisfy the drinking water quality requirements.

Point 4: (line40) those are not problems on sustainable development, they are the cause of it. Please clarify it. 

Response 4: We change “problem” to “challenges”. The logic here is that water pollution and air pollution are very serious and they threaten the realization of sustainable development. Therefore, “challenges” is better than “problem”or “cause”.

Point 5: (line41-44) these two sentences are controversial to each other, please clarify.

Response 5: Here, the two sentences are actually not controversial to each other. Rather, it presents two reasons why households eagerly hope to reduce pollution and improve the environment. The first sentence states that pollution causes great economic costs to households (so they wish to reduce pollution). The second sentence presents a more complicate story: in the past, when living standards were very low, the essential need is economic development; when economic conditions are improved, households now expect better quality of environment as well even at the expense of some economic benefits (so they wish to reduce pollution). To make the second sentence clearer, we add “economic conditions”. The new version is :

On the one hand, environmental pollution severely threatens people's physical and mental health and carries enormous economic costs. On the other hand, with improvements in economic conditions and living standards, households now expect higher quality of health and environment, and are even willing to pay for this.

Point 6: (line49) and what are that research?

Response 4: Existing research [2-4] has recognized the important role of the local governments' land resource allocation behavior on environmental problems.

  1. Liu, Y.; Chen, J. Land System, Financing Mode and Industrilization with Chinese Characteristics. China Industrial Economics, 2020, (03).
  2. Liang, P.; Gao, N. Personnel Change, Legal Environment and Local Environmental Pollution. Management World, 2014, (06).
  3. Long, S.; Hu, J. On Environmental Pollution from the Perspective of Government-enterprise Collusion: Theoretical and Empirical Analysis. Journal of Finance and Economics, 2014, 40, (10).

Point 7: (line100-101) Canada belongs to North America, and Spain not a region (write South-West Europe)

Response 7: Change “Many other empirical research proved the validity of EKC in different regions of the world, including North America [6], Canada [7], and Spain [8].” to ”Many other empirical research proved the validity of EKC in different regions of the world, including North America [6,7], and South-West Europe [8]

Point 8: (Figure 1) Why is the legend in bold?

Response 8: In Figure 1, the words in the boxes are set in bold because we hope that the readers could have a clearer view of the figure. The “Figure 1” in the legend is bold based on the requirement of the template.

Point 9: (Table 2) please indicate the meaning of the abbreviations

Response 9: We change ”S.D.” to “Standard Deviation”.

Point 10: (line300-317) it is a result – not the data source -, please move this part into the result section

Response 10: Thank you for your advice here. We agree with your comment that the summary statistics here is not a “data source”. However, we afraid that it is also not very appropriate to move it to “results” as they are only some data collected then just been simply treated and crudely reported. After discussing with all our authors, we decided to change the title of section 3.2 from data sources to data.

Point 11: What parts of China considered to be the given 3 regions? Please indicate the exact borders!

Response 11: Thank you so much for pointing out the missing region dividing here. We include a new Figure to indicate the West - Middle - East regions in section.

4.1. Spatial distribution of PM2.5 and industrial land supply

In 1986, "seven five" plan divided mainland Chinese economic area into east-region, middle-region and west-region. Here, we provide a figure for readers to have a general understanding of Chinas official division of regions.

Figure 3. Division of Western, Central and Eastern China

Point 12: (Figure 3) too small, I have got good eyes, but I cannot read the years included on the separate figures. (a) is 2011? and (b) is 2009? if I can read it well, why these are not in order? (Figure 4) subfigures are not in the correct place, same stands here- too small.

Response 12: In Figure 3, we felt very sorry to make a mistake here. Subfigure 3(a) and 3(b) were reversely placed in the original version and we re-ordered them in the revised version to satisfy a correct time order.

For clear views of the figures, we also enlarged the size of original Figure 3 and 4. Also, since we add a new Figure 3, the original Figure 3 and 4 becomes Figure 4 and 5 in the revised version. Please see the new manuscript file for viewing these figures.

Point 13: Introduction and Literature review: I do not feel that these need to be a separate chapter. Please cover both in just the Introduction part.

Response 13: The introduction and literature review here contains different contents and serve different functions.

The introduction section first introduces the research background. Then it states the basis, aim and scope of the research. Lastly it provides a brief organization of the following sections.

The literature review section mainly discusses existing researches relating to environmental pollution and its influential factors (paragraph 1), local governments’ land supply behavior (paragraph 2 and 3) and impact of industrial land supply on pollution (paragraph 4). After analyzing the gap of existing literature, we provide a hypothesized conceptual model which actually reflects most literature that we discussed in this section.

In fact, these two parts are correlated but not the same. Many other research articles also contains both sections. After serious consideration, we decided to remain both parts.

Other Changes:

  1. The introduction section first introduces the research background. Then it states the basis, aim and scope of the research. Lastly it provides a brief organization of the following sections.The second paragraph contains some redundant statement, so we deleted some part of it:

Existing research has recognized the important role of the local governments' land resource allocation behavior on environmental problems. In the context of fiscal decentralization and the "GDP-growth tournament among governments", local government officials are generally inclined to adopt the model of "developing through land" due to financial and political career promotion incentives [2]. They tend to pursue regional economic growth goals and maximize fiscal revenue without fully considering regional environmental protection and governance. Specifically, local governments tend to adopt the method of transferring industrial land by agreement and apply a low entry threshold for absorbing investment to attract industrial enterprises to the region and  relax regional environmental regulations which aggravates environmental pollution. In addition, the extensive development model of "land for capital," selling industrial land cheaply, can quickly increase regional fixed asset investment, cultivate new tax sources, make up the gap between fiscal revenuw and expenditure, and improve economic growth. With the advantage of a "two-way monopoly" in land transactions supported by China's land property rights system [3], local governments expropriate land from rural areas at a price lower than the land's value, and then supply land for real estate, commerce, and service industries at a high price to gain considerable land transfer fees. This differentiated land allocation strategy for industrial, commercial, and residential land is local governments' optimal strategic choice to maximize their own utility. However, the resulting inefficient supply of industrial land and damage to the ecological environment needs to be carefully evaluated, but it has not been to date.

  1. In order to emphasize the main method that we use in this article, we change the title from A Quantitative Estimation on the Impact of Industrial Land Supply on Environmental Pollution in Urban China from 2009 to 2017to A Two-way Fixed Effect Estimation on the Impact of Industrial Land Supply on Environmental Pollution in Urban China.

There are also some other revisions and grammar corrections in our revised submission and all changes can be traced and viewed in our new MS Word Template.

Yours sincerely

The Authors

Round 2

Reviewer 2 Report (New Reviewer)

Dear Authors!

Thank you for detailing and making all of the suggested modifications. Now, with all of the changes it stands as a good and scientifically fresh type of research.

In this form, I can accept the modifications.

This manuscript is a resubmission of an earlier submission. The following is a list of the peer review reports and author responses from that submission.

Round 1

Reviewer 1 Report

I have reviewed your article “A Two-way Fixed Effect Estimation on the Impact of Industrial Land Supply on Environmental Pollution in Urban China” and have suggestions or questions for the sections and a series of edits following that.

Line 220: unless this is the journal style, why are the results explaining the methods? Are the methods derived from the literature review in section 2? This is confusing. Methods should be explained in a methods section.

Sections 3.1 and 3.2 are too long; get to the hypothesis faster or state the hypotheses first and establish some rationale or refer to the literature review.

A study with this kind of data use from multiple databases including methods for dealing with missing data needs a limitation section (typically a paragraph). Lines 621 and 622 are not enough.

Section 5: please shorten the conclusions.

Overall, this was an interesting paper. However, I feel that it could be reduced in word length, especially the review of the literature and the conclusions. There seems to be quite a bit of unnecessary repetition throughout the paper. Also, what is the point of this paper in plain language? I am unsure what a 0.0623% increase translates to in real life — more mortality, illness? Please elaborate

Grammatical edits:

Line 34: add an “s” to standard

Line 34: what is the qualified proportion of surface water? It seems to reference something to do with pollution; please clarify for readers.

Line 55: synonym for aggravate?

Line 60: add an “s” to the end of resource.

Line 67: but is has not been to date

Line 89: it not tt

Line 92: “reduce housing prices”

Line 93: evidence not evidences

Line 112-113: that an inverted U-shaped relationship exists.

Line 114: the western region demonstrates

Line 131: finance not fiance

Line 134: replaced reduce with reduced 

Line 155: it is not clear what is being said here

Line 159: please reword this sentence to shorten it and make it clear

Lines 178-179: add references as examples of this existing literature.

Lines 185-189: synthesize a bit better, it’s too wordy and from the same reference, so shorten it into a single long sentence or two smaller sentences.

Line 282: how many (n) for the missing values? This should be clear and exact what was carried out to replace missing values.

Line 292: what other pollutants? The ones mentioned earlier or new ones?

Section 4; results? Why empirical results?

Line 322: provide

Line 329: space between deep red and parenthesis

Line 354: references examples?

Line 365: two-year lag is mentioned twice; unclear what the difference is

Line 405: space between PM2.5 and (ug/m^3) - likely only need to mention units once

Line 428: same as line 405

Line 527: same as line 405

Line 529: same as line 405

Author Response

Dear Reviewer,

Thank you so much for your kindly reply and helpful suggestions. We really appreciate your encouragement and advice. During the past week, we have been trying to revise our work based on your suggestions. Here are some main changes to be noticed (you could see the attached word file to avoid format errors):

Point 1: Line 220: unless this is the journal style, why are the results explaining the methods? Are the methods derived from the literature review in section 2? This is confusing. Methods should be explained in a methods section.

Response 1: Section 3 name are typed wrongly in the last version of our article. As you have pointed out, it should not be “Results”. Instead, it is the “Methodology” section. We sincerely apologize for such a careless mistake.

To make clearer introduction of the method that we use in this article, we enriched the content by a) including more explanations about the main method (two-way fixed effect model) and its benefits, and b) summarize all the methods that we use in this article (descriptive analysis, benchmark analysis, robustness check, regional heterogeneity check, and mechanism research) at the beginning of Section 3.2 (Methods and data) from Line 241 to Line 272:

“3.2. Methods and data

To investigates the impacts of local governments’ land supply behavior on environmental pollution, we constructed a data set comprising observations for 277 Chinese cities from 2009 to 2017. Based on this data set, we apply the following methods: Firstly, to provide a clear insight of the national level distribution and regional differences, we use GIS to display the spatial distribution of pollution(annual average concentration of PM2.5) and industrial land supply respectively. By comparing the change of pollution with the ratio of industrial land supply, we could roughly deduce the qualitative impacts of land supply behavior on environmental pollution. Secondly, to quantitatively evaluate the impacts of land supply behavior, we use a two-way fixed effect regression model as our benchmark model. The two-way fixed effect model are now widely employed to estimate treatment effects. Chaisemartin and D’Haultfœuille found that from 2010 to 2012, 19% of empirical articles on American Economic Review(AER) applied the two-way fixed effect regression to evaluate the effect of a treatment on an outcome [36]. It could adjust for two types of unobserved confounders(unit-specific and time-specific) at the same time[37] by including both sets of fixed terms, namely cities and years. The city fixed effects describe the permanent differences between cities. The year fixed effects capture the impacts that are common to all cities but vary by year. This two-way fixed effect model is a useful, scientific and rigorous tool as it could provide more accurate econometric estimates and greatly alleviate the disturbance from multi-collinearity thus improve the reliability of our conclusions [38,39]. Thirdly, we test the robustness through two ways: use other pollutants such as PM10 and CO2 to replace the PM2.5 concentration; exclude the four municipalities, provincial capital cities and sub-provincial cities from our original sample. Fourthly, in order to examine the differences between regions, we further divide the sample into eastern, central, and western regions and perform group-level regressions. So we can further investigate the regional heterogeneity. Lastly, we run the mechanism analysis from three dimensions: industrial structure, industrial development level, and local finance. Specifically, we first apply the classic two-step method to test the intermediary variables, then run the Sobel test and bootstrap test for the intermediary variables that do not meet the significant conditions of the two-step method. ” 

Point 2: Sections 3.1 and 3.2 are too long; get to the hypothesis faster or state the hypotheses first and establish some rationale or refer to the literature review.

Response 2: After discussion and careful consideration, we shorten the Section 3.1 (Hypotheses) and attempted to get the two hypotheses faster by referring to contents in Literature Review. Our revised Section 3.1 are from Line 231 to 240:

“3.1. Hypotheses

Based on the above framework, we formalize the following hypothesis:

Hypothesis1.Increased supply of industrial land by local government has a negative impact on regional environmental conditions.

From the discuss in the previous section, we konw that in different regions, local government land quota allocations have different environmental impacts. It is therefore necessary to investigate the effect on the environment of different regional approaches to land transfer. Thus, we propose another hypothesis:

Hypothesis2.Increased supply of industrial land by local government has heterogeneous effects on environmental conditions in different regions.”

Point 3: A study with this kind of data use from multiple databases including methods for dealing with missing data needs a limitation section (typically a paragraph). Lines 621 and 622 are not enough.

Response 3: In our previous version, a concentrated expression of limitations were missing. After revising, we put all limitations together in the last paragraph and add one new point (assumption of linear additive effects). Line 603 to Line 617 are our new limitations section:

We acknowledge the limitations of our study. First, we use a two-way fixed effect model in our article to simultaneously adjust for cities and years fixed effects. However, whether these two types of unobserved confounders can be adjusted at the same time relies on the assumption of linear additive effects [37], which may need further examination. In addition, we only investigate the impact of industrial land supply and aggregate all other land use types as “others”. Future researchers could do more analysis on other different land use types, such as residential and commercial land respectively, to explore if these types of land supply have certain impacts on environmental pollution. In addition, the formation and spread of environmental pollution is not limited to the city in the administrative sense, there are often spillover effects of environmental pollution between regions. Whether land transfer competition between regions will deepen mismatches of land resources and thus aggravate environmental pollution, and whether there are other mechanisms by which local government land transfer affects environmental pollution in the jurisdiction area, are topics that deserve further study.

Point 4: Section 5: please shorten the conclusions.

Response 4: We revised our main contribution part in the conclusion Section by expressing more concisely the main results in previous sections. Our contribution section in the revised manuscript are from Line 592 to Line 602:

“This study's main contributions are as follows. First, based on industrial land supply data for prefecture-level cities from the National Land and Resources Yearbook, this paper describes the behavior of local governments in industrial land selling more accurately and comprehensively than prior studies, and explains regional environmental pollution from this perspective. Second, this paper considers the effects of government officials' promotion incentives and interactions between local governments. We explore the environmental externalities brought about by differences in local governments' land transfer behaviors from the perspective of land resource allocation and provide empirical evidence for how the institutional design of regional land resource allocation can be improved. Finally, this paper analyzes the mechanisms behind the environmental externalities caused by local governments' land allocation behavior. “

Point 5: Overall, this was an interesting paper. However, I feel that it could be reduced in word length, especially the review of the literature and the conclusions. There seems to be quite a bit of unnecessary repetition throughout the paper. Also, what is the point of this paper in plain language? I am unsure what a 0.0623% increase translates to in real life — more mortality, illness? Please elaborate

Response 5: We carefully reduced some wordy part in the literature review section and conclusions. We re-arrange Section 2 (Literature Review) to make it more organized. In order to avoid repetition from Section, we deleted some pieces of literature in this part and shorten some of the paraphrases. Line 87 to Line 179 are the remaining part from last version of literature review after reduction:

“Environmental pollution and its influential factors have been important topics of research among scholars around the world. Many researchers attempt to explain the evolution of environmental conditions from the perspective of economic development.  Grossman and Krueger [4] found that concentrations of sulfur dioxide and smoke initially increase with per capita GDP at low levels of national income, then decrease with higher per capita GDP, and finally level off with per capita GDP of US$8000 (1985 dollars). This inverted U-shaped relationship is commonly referred as environmental Kuznets curve (EKC). Hettige et al. found a similar inverted U-shaped relationship between GDP and the toxic intensity of manufacturing industries [5]. Many other empirical research proved the validity of EKC in different regions of the world, including North America [6], Canada [7], and Spain [8]. Some research found similar results in China. For example, Wang and Huang [9] found a U-shaped relationship between economic growth and air pollution in 112 cities in China. Using an ARDL model, Akram et al. [10] found strong evidence of the EKC for China. Shao et al. [11] found a statistically significant U-shaped relationship between the degree of smog pollution and regional economic growth, but that most of the eastern region is still on the left side of the EKC; that is, the environment is gradually deteriorating with economic development. However, some researchers do not agree with the shape of the EKC. Xu and Song [12] found an inverted U-shaped relationship exists between income and pollution in China as a whole and in the eastern and central regions, although the western region demonstrates a U-shaped pattern. A condition for establishing the EKC's inverted U-shaped curve is that when economic development reaches a certain stage, measures such as increased investment in environmental protection, improved quality of governance [13], technological progress, improved energy efficiency [14], and the relocation to different places of high-polluting and high-energy industries can curb or alleviate environmental pollution. However, if institutional barriers inhibit the above mechanisms, then a higher level of economic development may not be accompanied by reduced environmental pollution. Therefore, the problem of environmental pollution is partly rooted in the administrative management system, so it is necessary to explore the role of the local governments in environmental pollution.

Under China's system of decentralization, local governments implement environmental protection policies promulgated by the central government, and are also the bridge between the central government and local enterprises [15]. As the designers and implementers of regional policies, local governments' actions will inevitably affect economic development and environmental conditions in their jurisdictions, which are reflected in the allocation of land resources and discretionary decisions in implementing policies [16]. For a long time, local governments in China have been more concerned about economic growth than environmental pollution. Local officials usually have insufficient incentive to implement environmental supervisions [17] due to their political career promotion incentives, and will tend to allocate resources to areas that can promote short-term economic growth. The motivations for local governments' land transfer behaviors are also inseparable from the financial and political promotion incentives for officials. On one hand, land finance is an important source of income for local governments in China. On the other hand, local officials are motivated to "collaborate" with companies to achieve mutual benefits: the local governments earn large profits from land transfer agreements, and reduced environmental regulations and supervision to support high pollution and high energy consumption enterprises; these enterprises attract investment and create long-term benefits such as fixed asset investment, employment, attraction of talented workers, and financial taxation for the local governments. In 2013, the central government incorporated local environmental protection indicators into the official performance assessment system. However, due to the positive externalities of environmental governance and information asymmetry in the top-down assessment system of central and local principal-agent systems, local government officials still have insufficient incentives for environmental protection. The existing literature argues that "government-enterprise collusion" is one of the reasons for the high level of environmental pollution in China [18,19]. Guo and Yao [20] argued that local governments have neglected environmental protection and pursued high economic growth jointly with enterprises, resulting in serious water pollution problems. The behavior of local governments is closely related to their jurisdictions' environmental conditions, and it is necessary to incorporate the behavior of local governments into the research framework for environmental issues.

Through strategic interactions, local governments fiercely compete in attracting investment and realizing economic growth [21,22]. In order to attract high-quality projects, local governments compete to reduce environmental regulatory standards, increase the scale of industrial land transfers and the proportion of land supplied by agreement, and introduce bottom-line competition for low-quality investment. Local governments that are geographically adjacent or have similar levels of economic development will also compete to imitate each other's land supply behavior. Fiscal incentives and land mismatches have resulted in a large supply of new construction land, which has led to excessive consumption of planned land quotas [23]. The central government promulgated the State Council's Notice on Strengthening Land Control and Related Issues on August 31, 2006, which stipulated that from January 1, 2007, industrial land must be sold by auction, listing, or open tender. However, the proportion of land transfers made by local government agreements remains relatively high, and the "race to the bottom" between local governments explains this phenomenon. In addition, the differentiated transfer strategies for different types of land are rational under financial incentives and external constraints [24], but they can easily lead to improper allocation of land resources, resulting in misallocation of land resources in industrial and service sectors [25]. Further, misallocation of land resources has been found to have a negative impact on allocation of financial lending [26], thereby adversely affecting both economic development and environmental protection. Most of the existing literature asserts that transfers of industrial land through agreements introduce relatively low-quality investment[27,28] and cause lower output efficiency[29,30] which result in more serious the pollution. The method and pricing of local government industrial land supplies will affect the economic performance of regions and enterprises. Therefore, improving the method of allocating land resources can increase enterprise efficiency and promote regional economic and environmental development.

Although many scholars have broadly studied pollution and industrial land supply behavior by local governments respectively, little research has focused on the impact of industrial land supply on pollution [31-34]. However, due to data limitations and the heterogeneity of plot endowments, few existing literature discuss the rationales behind local governments' industrial land supply behavior and its impact on environmental pollution in urban China.”

0.0623% means that: if the industrial land ratio increased by 1%, PM2.5 concentration will increase 0.0623 percentage. For an average level of lpm2.5 at 3.71 (40.85μg/m3), it will increase to 3.7723(43.48μg/m3) which represents an increase of 0.133 standard deviation.

We are also grateful that you pointed out some of our grammar mistakes. Here are some of our corrections:

Line 35: “Standard” to “Standards”

Line 36: “the qualified proportion of national surface water is 83.4%” to “83.4% of the national surface water sections are qualified”

Line 61: “resource” to “resources”

Line 69: “which has not been” to “but it has not been to date”

Line 105: “there does exist an inverted U-shaped relationship” to “an inverted U-shaped relationship exists”

Line 107: “demonstrate” to “demonstrates”

Line 129: “land fiance” to “land finance”

Line 132: “reduce” to “reduced”

Line 167 to Line 170: adding references and reducing wordy expressions, “Most of the existing literature asserts that transfers of industrial land through agreements introduce relatively low-quality investment[27,28] and cause lower output efficiency[29,30] which result in more serious the pollution.”

Line 311: “Empirical results” to “Results”

Line 361: the second “two year lag” changes to “both lags”

There are also some other revisions in our revised submission and all changes can be viewed in our new MS Word Template. We appreciate all your time and effort spent.

Wish you a nice new week!

Yours sincerely

Yan Xiangqi, Tuo Hanbing and Lai Yani

Reviewer 2 Report

The topic of this paper is very meaningful, but the research is not in-depth, and the comprehensive evaluation suggests that the manuscript be rejected. The main reasons are as follows:1. The literature review of this article is too long, and the key points are not put forward;2. The research method in this paper is too simple and does not solve the problem well;3. The research data in this paper is too old to be published.The author can add effective research methods, supplement data, and resubmit the article after making major revisions.

Author Response

Dear Reviewer,

Thank you so much for your kindly reply and advice. We appreciate your time and effort spent. During the past week, we have been trying to revise our work based on peer review comments. Here are some main changes to be noticed (To avoid format error, we attached the response file for you to read):

Point 1: The topic of this paper is very meaningful, but the research is not in-depth, and the comprehensive evaluation suggests that the manuscript be rejected. 

Response 1: We feel grateful for your sincere opinion. However, we do not agree with your comment that this research is not in-depth. In our research, we find that although many scholars have broadly studied pollution and industrial land supply behavior by local governments respectively, little research has focused on the impact of industrial land supply on pollution. Due to data limitations and the heterogeneity of plot endowments, few existing literature discuss the rationales behind local governments' industrial land supply behavior and its impact on environmental pollution in urban China. Based on existing literature and our hypothesized framework, we proposed two main hypotheses. To quantitatively evaluate the impacts of land supply behavior, we use a two-way fixed effect regression model as our benchmark model. Then we test the robustness through two ways. Then, we further investigate the regional heterogeneity. Lastly, we run the mechanism analysis from three dimensions: industrial structure, industrial development level, and local finance. We find that: (1) When local governments increase the proportion of industrial land, this will aggravate regional environmental pollution, with its main impact on forms of air pollution. (2) The impact of industrial land supply on environmental pollution shows significant regional heterogeneity. The positive impact of the proportion of industrial land on environmental pollution is stronger in the central and western regions. (3) Greater industrial land transfer by local governments leads to expansion of regional secondary industry and increases in local fiscal deficit. Unbalanced industrial development, insufficient corporate innovation, and insufficient investment in environmental protection will increase pollution.

Point 2: The literature review of this article is too long, and the key points are not put forward

Response 2: We carefully reduced some wordy part in the literature review section and conclusions. We re-arrange Section 2 (Literature Review) to make it more organized. In order to avoid repetition from Section, we deleted some pieces of literature in this part and shorten some of the paraphrases. Line 87 to Line 179 are the remaining part from last version of literature review after reduction:

“Environmental pollution and its influential factors have been important topics of research among scholars around the world. Many researchers attempt to explain the evolution of environmental conditions from the perspective of economic development.  Grossman and Krueger [4] found that concentrations of sulfur dioxide and smoke initially increase with per capita GDP at low levels of national income, then decrease with higher per capita GDP, and finally level off with per capita GDP of US$8000 (1985 dollars). This inverted U-shaped relationship is commonly referred as environmental Kuznets curve (EKC). Hettige et al. found a similar inverted U-shaped relationship between GDP and the toxic intensity of manufacturing industries [5]. Many other empirical research proved the validity of EKC in different regions of the world, including North America [6], Canada [7], and Spain [8]. Some research found similar results in China. For example, Wang and Huang [9] found a U-shaped relationship between economic growth and air pollution in 112 cities in China. Using an ARDL model, Akram et al. [10] found strong evidence of the EKC for China. Shao et al. [11] found a statistically significant U-shaped relationship between the degree of smog pollution and regional economic growth, but that most of the eastern region is still on the left side of the EKC; that is, the environment is gradually deteriorating with economic development. However, some researchers do not agree with the shape of the EKC. Xu and Song [12] found an inverted U-shaped relationship exists between income and pollution in China as a whole and in the eastern and central regions, although the western region demonstrates a U-shaped pattern. A condition for establishing the EKC's inverted U-shaped curve is that when economic development reaches a certain stage, measures such as increased investment in environmental protection, improved quality of governance [13], technological progress, improved energy efficiency [14], and the relocation to different places of high-polluting and high-energy industries can curb or alleviate environmental pollution. However, if institutional barriers inhibit the above mechanisms, then a higher level of economic development may not be accompanied by reduced environmental pollution. Therefore, the problem of environmental pollution is partly rooted in the administrative management system, so it is necessary to explore the role of the local governments in environmental pollution.

Under China's system of decentralization, local governments implement environmental protection policies promulgated by the central government, and are also the bridge between the central government and local enterprises [15]. As the designers and implementers of regional policies, local governments' actions will inevitably affect economic development and environmental conditions in their jurisdictions, which are reflected in the allocation of land resources and discretionary decisions in implementing policies [16]. For a long time, local governments in China have been more concerned about economic growth than environmental pollution. Local officials usually have insufficient incentive to implement environmental supervisions [17] due to their political career promotion incentives, and will tend to allocate resources to areas that can promote short-term economic growth. The motivations for local governments' land transfer behaviors are also inseparable from the financial and political promotion incentives for officials. On one hand, land finance is an important source of income for local governments in China. On the other hand, local officials are motivated to "collaborate" with companies to achieve mutual benefits: the local governments earn large profits from land transfer agreements, and reduced environmental regulations and supervision to support high pollution and high energy consumption enterprises; these enterprises attract investment and create long-term benefits such as fixed asset investment, employment, attraction of talented workers, and financial taxation for the local governments. In 2013, the central government incorporated local environmental protection indicators into the official performance assessment system. However, due to the positive externalities of environmental governance and information asymmetry in the top-down assessment system of central and local principal-agent systems, local government officials still have insufficient incentives for environmental protection. The existing literature argues that "government-enterprise collusion" is one of the reasons for the high level of environmental pollution in China [18,19]. Guo and Yao [20] argued that local governments have neglected environmental protection and pursued high economic growth jointly with enterprises, resulting in serious water pollution problems. The behavior of local governments is closely related to their jurisdictions' environmental conditions, and it is necessary to incorporate the behavior of local governments into the research framework for environmental issues.

Through strategic interactions, local governments fiercely compete in attracting investment and realizing economic growth [21,22]. In order to attract high-quality projects, local governments compete to reduce environmental regulatory standards, increase the scale of industrial land transfers and the proportion of land supplied by agreement, and introduce bottom-line competition for low-quality investment. Local governments that are geographically adjacent or have similar levels of economic development will also compete to imitate each other's land supply behavior. Fiscal incentives and land mismatches have resulted in a large supply of new construction land, which has led to excessive consumption of planned land quotas [23]. The central government promulgated the State Council's Notice on Strengthening Land Control and Related Issues on August 31, 2006, which stipulated that from January 1, 2007, industrial land must be sold by auction, listing, or open tender. However, the proportion of land transfers made by local government agreements remains relatively high, and the "race to the bottom" between local governments explains this phenomenon. In addition, the differentiated transfer strategies for different types of land are rational under financial incentives and external constraints [24], but they can easily lead to improper allocation of land resources, resulting in misallocation of land resources in industrial and service sectors [25]. Further, misallocation of land resources has been found to have a negative impact on allocation of financial lending [26], thereby adversely affecting both economic development and environmental protection. Most of the existing literature asserts that transfers of industrial land through agreements introduce relatively low-quality investment[27,28] and cause lower output efficiency[29,30] which result in more serious the pollution. The method and pricing of local government industrial land supplies will affect the economic performance of regions and enterprises. Therefore, improving the method of allocating land resources can increase enterprise efficiency and promote regional economic and environmental development.

Although many scholars have broadly studied pollution and industrial land supply behavior by local governments respectively, little research has focused on the impact of industrial land supply on pollution [31-34]. However, due to data limitations and the heterogeneity of plot endowments, few existing literature discuss the rationales behind local governments' industrial land supply behavior and its impact on environmental pollution in urban China.”

Point 3: The research method in this paper is too simple and does not solve the problem well. 

Response 3: We do not think that our research method is too simple. Rather, we apologize for not clearly describing our methods in the previous version of manuscript. To make clearer presentation of the method that we use throughout our research, we enriched the content by a) including more explanations about the main method (two-way fixed effect model) and its benefits, and b) summarize all the methods that we use in this article (descriptive analysis, benchmark analysis, robustness check, regional heterogeneity check, and mechanism research) at the beginning of Section 3.2 (Methods and data) from Line 241 to Line 272:

“3.2. Methods and data

To investigates the impacts of local governments’ land supply behavior on environmental pollution, we constructed a data set comprising observations for 277 Chinese cities from 2009 to 2017. Based on this data set, we apply the following methods: Firstly, to provide a clear insight of the national level distribution and regional differences, we use GIS to display the spatial distribution of pollution(annual average concentration of PM2.5) and industrial land supply respectively. By comparing the change of pollution with the ratio of industrial land supply, we could roughly deduce the qualitative impacts of land supply behavior on environmental pollution. Secondly, to quantitatively evaluate the impacts of land supply behavior, we use a two-way fixed effect regression model as our benchmark model. The two-way fixed effect model are now widely employed to estimate treatment effects. Chaisemartin and D’Haultfœuille found that from 2010 to 2012, 19% of empirical articles on American Economic Review(AER) applied the two-way fixed effect regression to evaluate the effect of a treatment on an outcome [36]. It could adjust for two types of unobserved confounders(unit-specific and time-specific) at the same time[37] by including both sets of fixed terms, namely cities and years. The city fixed effects describe the permanent differences between cities. The year fixed effects capture the impacts that are common to all cities but vary by year. This two-way fixed effect model is a useful, scientific and rigorous tool as it could provide more accurate econometric estimates and greatly alleviate the disturbance from multi-collinearity thus improve the reliability of our conclusions [38,39]. Thirdly, we test the robustness through two ways: use other pollutants such as PM10 and CO2 to replace the PM2.5 concentration; exclude the four municipalities, provincial capital cities and sub-provincial cities from our original sample. Fourthly, in order to examine the differences between regions, we further divide the sample into eastern, central, and western regions and perform group-level regressions. So we can further investigate the regional heterogeneity. Lastly, we run the mechanism analysis from three dimensions: industrial structure, industrial development level, and local finance. Specifically, we first apply the classic two-step method to test the intermediary variables, then run the Sobel test and bootstrap test for the intermediary variables that do not meet the significant conditions of the two-step method. ” 

Point 4: The research data in this paper is too old to be published.

Response 4: The main purpose of this article is to investigate the impacts of local governments’ land supply behavior on environmental pollution. Our data set comprises observations for 277 Chinese cities from 2009 to 2017. It comes from multiple data sets and includes industrial land supply data, city data and environment data. It meets the requirement for serving our main purpose and solve our problem.

Besides, we also made some other major changes:

  1. After discussion and careful consideration, we shorten the Section 3.1 (Hypotheses) and attempted to get the two hypotheses faster by referring to contents in Literature Review. Our revised Section 3.1 are from Line 231 to 240:

“3.1. Hypotheses

Based on the above framework, we formalize the following hypothesis:

Hypothesis1.Increased supply of industrial land by local government has a negative impact on regional environmental conditions.

From the discuss in the previous section, we konw that in different regions, local government land quota allocations have different environmental impacts. It is therefore necessary to investigate the effect on the environment of different regional approaches to land transfer. Thus, we propose another hypothesis:

Hypothesis2.Increased supply of industrial land by local government has heterogeneous effects on environmental conditions in different regions.”

  1. In our previous version, a concentrated expression of limitations were missing. After revising, we put all limitations together in the last paragraph and add one new point (assumption of linear additive effects). Line 603 to Line 617 are our new limitations section:

“We acknowledge the limitations of our study. First, we use a two-way fixed effect model in our article to simultaneously adjust for cities and years fixed effects. However, whether these two types of unobserved confounders can be adjusted at the same time relies on the assumption of linear additive effects [37], which may need further examination. In addition, we only investigate the impact of industrial land supply and aggregate all other land use types as “others”. Future researchers could do more analysis on other different land use types, such as residential and commercial land respectively, to explore if these types of land supply have certain impacts on environmental pollution. In addition, the formation and spread of environmental pollution is not limited to the city in the administrative sense, there are often spillover effects of environmental pollution between regions. Whether land transfer competition between regions will deepen mismatches of land resources and thus aggravate environmental pollution, and whether there are other mechanisms by which local government land transfer affects environmental pollution in the jurisdiction area, are topics that deserve further study.”

  1. We revised our main contribution part in the conclusion Section by expressing more concisely the main results in previous sections. Our contribution section in the revised manuscript are from Line 592 to Line 602:

“This study's main contributions are as follows. First, based on industrial land supply data for prefecture-level cities from the National Land and Resources Yearbook, this paper describes the behavior of local governments in industrial land selling more accurately and comprehensively than prior studies, and explains regional environmental pollution from this perspective. Second, this paper considers the effects of government officials' promotion incentives and interactions between local governments. We explore the environmental externalities brought about by differences in local governments' land transfer behaviors from the perspective of land resource allocation and provide empirical evidence for how the institutional design of regional land resource allocation can be improved. Finally, this paper analyzes the mechanisms behind the environmental externalities caused by local governments' land allocation behavior. “

We also correct some of our grammar mistakes. Here are some of our corrections:

Line 35: “Standard” to “Standards”

Line 36: “the qualified proportion of national surface water is 83.4%” to “83.4% of the national surface water sections are qualified”

Line 61: “resource” to “resources”

Line 69: “which has not been” to “but it has not been to date”

Line 105: “there does exist an inverted U-shaped relationship” to “an inverted U-shaped relationship exists”

Line 107: “demonstrate” to “demonstrates”

Line 129: “land fiance” to “land finance”

Line 132: “reduce” to “reduced”

Line 167 to Line 170: adding references and reducing wordy expressions, “Most of the existing literature asserts that transfers of industrial land through agreements introduce relatively low-quality investment[27,28] and cause lower output efficiency[29,30] which result in more serious the pollution.”

Line 311: “Empirical results” to “Results”

Line 361: the second “two year lag” changes to “both lags”

There are also some other revisions in our revised submission and all changes can be traced and viewed in our new MS Word Template. Again, we appreciate all your time and effort spent.

Wish you a nice new week!

Yours sincerely

Yan Xiangqi, Tuo Hanbing and Lai Yani

Round 2

Reviewer 1 Report

Thank you for engaging with my comments and suggestions.

Please have a native English speaker read the final draft of the work as there are still areas with incorrect grammar (please watch out for correct use of pluralities).

I still do not understand what this manuscript is attempting to say in the "big picture?" How do your results translate into the real world? Will more people get sick or die from air pollution? Give readers an idea of what 0.0623% translates to in real life; it's such a small number and looks insignificant in the grand scheme. It is possible to allude to this in a discussion without implying causation. Do not rely on statistical significance so much to explain or "sell" the result to readers.

Reviewer 2 Report

This article has only made some simple deletions without adding effective modifications. The research method of this paper is simple and insufficient. I still refuse to comment!